# scRepli-RamDA-seq: a multi-omics technology enabling the analysis of gene expression dynamics during S-phase

Rawin Poonperm [1,2,6], Taiki Yoneda[2,6], Taito Imada[2], Saori Takahashi [1], Takako Ichinose[1], Hisashi Miura [1], Tetsutaro Hayashi [3], Mariko Kuse [3], Mika Yoshimura [3], Koji Nagao [4], Chikashi Obuse[4], Itoshi Nikaido [3,5] ✉, Ichiro Hiratani [1] ✉ & Shin-ichiro Takebayashi [2] ✉

Single-cell sequencing has advanced our understanding of cell-type diversity and heterogeneity. However, existing single-cell multi-omics methods lack the ability to monitor gene expression dynamics during S-phase progression. Here, we introduce single-cell (sc)Repli-RamDA-seq (scRR-seq), a multi-omics method that enables high-resolution DNA replication profiling and full-length total RNA sequencing from the same single cell in a haplotype-specific manner. scRR-seq generates DNA replication and RNA sequencing data comparable to individually obtained scRepli-seq and scRamDA-seq data, respectively. Unlike other scDNA/RNA-seq methods, scRR-seq allows one to tell the S-phase stage of a given cell based on the percentage of the replicated genome derived from scRepli-seq data. This facilitates analysis of gene expression dynamics during S-phase progression, enabling the identification of S-phase progression markers. scRR-seq also detects copy-number variation in non-S-phase cells and outperforms other scDNA/RNA-seq methods in various measures. Taken together, scRR-seq is a robust single-cell multi-omics method with promising potential for comprehensive genome/transcriptome analysis.

A number of single-cell sequencing methods have been developed that simultaneously profile DNA and RNA from the same cell[1-11]. Despite the advancements in methodology, improvement in the resolution and sensitivity of both DNA sequencing (DNA-seq) and RNA sequencing (RNA-seq) components has been limited. Most methods typically use the DNA-seq component to detect DNA copy-number variation (CNV) at a relatively low resolution of 0.25–1.0 Mb, with few utilizing DNA-seq data for detecting DNA sequence variants. This limitation is influenced by several factors, including the low genome coverage, possibly due to sample loss during the experiment, biases introduced during whole genome amplification (WGA), and/or insufficient sequencing depth[12]. Similarly, the RNA-seq component also has room for improvement. For instance, most current methods primarily capture polyadenylated [poly(A)] RNAs but not non-poly(A) RNAs, although the latter play important roles in various contexts. Moreover, current methods could detect roughly <10,000 genes per cell, which is lower than the number obtained by scRNA-seq performed individually. These issues highlight the need to upgrade methods to achieve higher resolution and broader applications for simultaneously profiling DNA and RNA from the same single cell.

[1]Laboratory for Developmental Epigenetics, RIKEN Center for Biosystems Dynamics Research (BDR), Kobe, Japan. [2]Laboratory of Molecular and Cellular Biology, Graduate School of Bioresources, Mie University, Tsu, Japan. [3]Omics AI Research Team, Advanced General Intelligence in Science Program (AGIS), TRIP Headquarters, RIKEN, Wako, Saitama, Japan. [4]Department of Biological Sciences, Graduate School of Science, The University of Osaka, Toyonaka, Japan. [5]Department of Functional Genome Informatics, Division of Biological Data Science, Medical Research Laboratory (MRL), Institute of Integrated Research (IIR), Institute of Science Tokyo, Tokyo, Japan. [6]These authors contributed equally: Rawin Poonperm, Taiki Yoneda. ✉e-mail: itoshi.nikaido@riken.jp; ichiro.hiratani@riken.jp; stake@bio.mie-u.ac.jp

Several years ago, we developed homemade single-cell DNA and RNA sequencing technologies, single-cell DNA replication sequencing (scRepli-seq)[13] and single-cell random displacement amplification sequencing (scRamDA-seq)[14], respectively. Our scRepli-seq represents one of the most advanced single-cell DNA-seq technologies[15], enabling genome-wide DNA replication analysis of single S-phase cells at a high resolution of ~40 kb and precise temporal mapping of S-phase progression of a given cell[13]. Importantly, because scRepli-seq is based on copy number calling, it can also detect CNVs in single cells[16]. scRamDA-seq represents one of the most advanced single-cell RNA-seq technologies, enabling full-length RNA sequencing at high sensitivity[14]. The ability of scRamDA-seq to detect intron reads with high sensitivity also makes it ideal for quantifying unspliced RNA and identifying recursive splicing at the single-cell level. Together, these methods provide powerful, standalone platforms for detailed single-cell DNA and RNA analysis.

Here, we report the development of a single-cell multi-omics method named scRepli-RamDA-seq (scRR-seq), which combines the strengths of scRepli-seq[13] and scRamDA-seq[14] to enable high-resolution DNA and RNA profiling from the same single cell. scRR-seq was made possible by the clean separation of genomic DNA and total RNA using magnetic beads. Importantly, scRR-seq provided data quality comparable to those obtained by performing scRepli-seq or scRamDA-seq individually. As a consequence, scRR-seq outperforms other scDNA/RNA-seq methods in both DNA-seq resolution and RNA-seq sensitivity. The ability of scRR-seq to precisely map the S-phase stage of a given cell allowed us to analyze gene expression dynamics during S-phase progression in an unprecedented manner and identify S-phase progression markers. scRR-seq also provided opportunities to study the intricate interplay between genomic instability and gene expression at the single-cell level. We describe the methodology of scRR-seq in detail and demonstrate its wide range of utility.

## Results

### The principle of the scRR-seq technology

To develop a single-cell DNA/RNA profiling method, we need to reliably separate DNA from RNA following single-cell isolation and cell lysis. To achieve this, we used the Dynabeads® MyOne™ Carboxylic Acid magnetic beads (Fig. 1a), as previous studies have shown that they can effectively separate DNA and RNA[6,10]. We further revealed that these beads could effectively capture genomic DNA from lysed cells but not naked DNA (Supplementary Fig. 1), suggesting that the beads captured chromatin rather than DNA per se, possibly through ionic bonds between their negatively charged carboxylic acid groups and chromatin proteins such as histones, which are highly positively charged due to abundant lysine/arginine residues. As a result, DNA is trapped on the beads, while RNA is released into the lysis buffer.

We subjected the isolated DNA on magnetic beads to WGA and next-generation sequencing (NGS) library preparation as in scRepli-seq[13]. In principle, scRepli-seq utilizes the degenerate oligonucleotide priming-PCR (DOP-PCR) technology, which provides relatively uniform WGA and good coverage[13,17]. The DNA-seq (scRepli-seq) component was used to assess DNA copy numbers genome-wide to karyotype G1-phase cells and analyze the replication state of S-phase cells. In parallel, the isolated RNA in the supernatant was subjected to scRamDA-seq[14]. This is achieved by using the not-so-random primers and the strand displacement amplification technique to amplify cDNA directly from the RNA template, which enables the detection of near full-length RNA, including non-poly(A) RNA, and results in higher detection rates of transcripts compared to other scRNA-seq methods[14].

We established two slightly different versions of the scRR-seq protocol, scRR1 and scRR3, using initial cell lysis buffer volumes of 1 μl and 3 μl, respectively. These two versions were developed in parallel, as each offers distinct advantages depending on the experimental context. scRR1 is more cost-effective and well-suited for large-scale experiments

where sample preparation is straightforward and a certain degree of failure is acceptable, such as when single cells are sorted by flow cytometry. In contrast, scRR3 is more appropriate for manually isolated samples that are difficult to collect, have larger carryover volumes, or are otherwise precious and require a high success rate (Supplementary Note 1, Supplementary Table 1, and Supplementary Fig. 2a).

After cell lysis, DNA on beads was processed similarly in both scRR1 and scRR3 based on the scRepli-seq protocol. In contrast, RNA in the lysis buffer was handled differently. For scRR3, we followed the protocol described in the kit (GenNext®RamDA-seq™ Single Cell Kit) with several modifications for the scRamDA-seq reaction. For scRR1, we used a smaller volume (1/3× of scRR3) for the scRamDA-seq reaction, which allowed us to skip the cDNA purification step and shorten the scRamDA-seq procedure (Supplementary Fig. 2a; see "Methods").

### scRR-seq generates high-resolution DNA replication timing data

To benchmark scRR-seq, we performed scRR1 and scRR3 using a near-diploid human cell line, hTERT-RPE1. We used fluorescence-activated cell sorting (FACS) to sort G1 and mid-S single cells into 0.2-ml tubes (Supplementary Fig. 2b) and processed them using our scRR-seq protocol. After filtering out cells that had failed the quality control (QC) test for library preparation (either DNA or RNA; see "Methods"), cells were subjected to NGS and downstream analysis. In addition, low-quality data were further excluded following the NGS analysis (see "Methods"; Supplementary Data 1).

We first analyzed the DNA-seq data from scRR-seq (scRR-seq-DNA hereafter) and compared them with our original scRepli-seq data obtained individually[13]. To construct replication timing (RT) profiles from scRR-seq-DNA, similar numbers of NGS reads per sample [~4 million (M) reads] were generated and analyzed using our existing analysis pipeline[13,18]. The number of reads mapped to the human reference genome before and after filtering was similar for all three methods tested (scRR1, scRR3, and scRepli-seq; Supplementary Fig. 3a). By analyzing scRR-seq-DNA from G1 cells, we could observe the known and expected CNV on the q-arm of chromosome 10 in hTERT-RPE1[19] (Supplementary Fig. 3b).

Because the WGA method for scRepli-seq provides relatively uniform amplification and sufficient coverage, we can assign replicated or unreplicated states to all genomic bins of S-phase cells after correcting mappability using control G1 cells[13]. We generated single-cell RT (scRT) profiles from scRR-seq-DNA for each S-phase cell at 40 or 80-kb resolution. In our hands, 40 kb is the highest resolution based on the optimal reads-per-bin analysis (Supplementary Fig. 3c). The scRT profiles derived from scRR-seq-DNA looked similar between 40 and 80-kb resolution data and were comparable to individually obtained scRepli-seq data (Fig. 1b and Supplementary Fig. 3d–f).

### scRR-seq generates high-quality RNA-seq data comparable to scRamDA-seq

Next, RNA-seq data from scRR-seq (scRR-seq-RNA hereafter) were analyzed and compared with scRamDA-seq data generated individually. Again, we obtained similar numbers of NGS reads per sample (~4 M reads) and analyzed them using the ramdaq analysis pipeline[20]. The number of reads mapped to the reference human genome was relatively similar for all methods (Supplementary Fig. 4a). We also confirmed that DNA/RNA cross-contamination is minimal in scRR-seq-RNA as in scRamDA-seq (Supplementary Fig. 4b). Correlation between External RNA Controls Consortium (ERCC) spike-in counts and their expected concentration were high ($R > 0.8$) and consistent in all samples (Supplementary Fig. 4c). The whole transcriptome-level comparison revealed a high correlation between scRR-seq-RNA and scRamDA-seq data ($R > 0.8$, Supplementary Fig. 4d). While both principal component analysis (PCA) and t-Distributed Stochastic Neighbor Embedding analyses revealed some degrees of differences across the three different protocols tested (Supplementary Fig. 4e, f), we observed

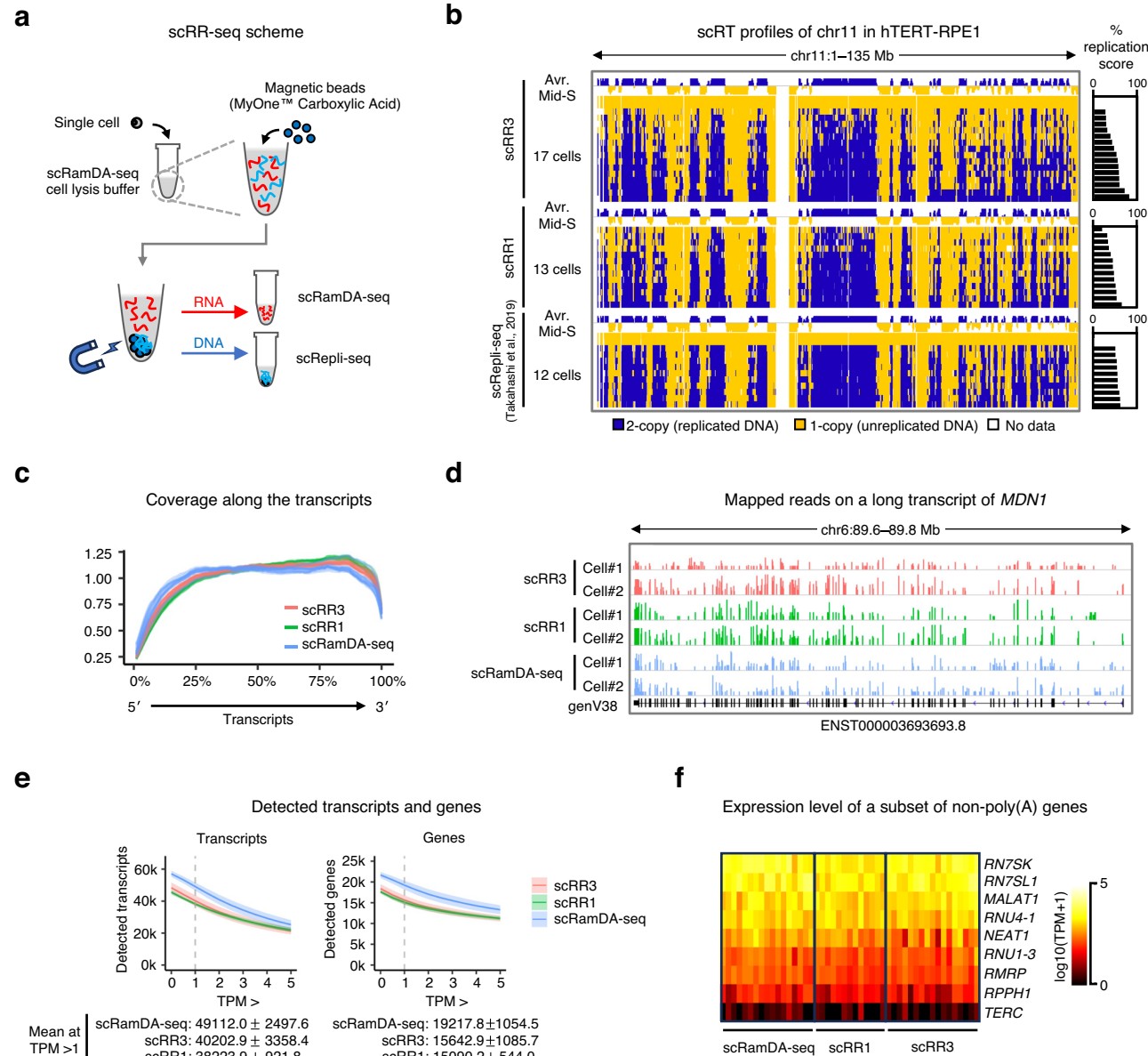

**Fig. 1 | scRR-seq enables scRepli-seq and scRamDA-seq from the same single cell. a** Schematic of the scRR-seq method. **b** Comparison of mid-S hTERT-RPE1 scRT profiles derived from scRR-seq (scRR3 and scRR1) and scRepli-seq. Chromosome 11 (chr11) is shown. The average mid-S RT profiles (Avr. Mid-S) are derived from cells with 40–70% replication scores. Yellow and blue represent unreplicated and replicated bins, respectively; white represents bins with no data. **c** Average of coverage along transcripts. **d** Comparison of scRR-seq and scRamDA-seq mapped reads on a representative long transcript, *MDN1* (~190 kb). **e** Detection rates of transcripts and genes expressed in mid-S-phase hTERT-RPE1 cells, across different expression level thresholds for each method. Here, transcripts refer to all RNA molecules transcribed from a given gene, including splicing variants, while genes are unique and their expression levels represent the sum of all transcripts assigned to each gene. The line and color-shaded areas represent means and standard deviations (SDs), respectively. The values at the bottom indicate the mean number of transcripts or genes with SDs at transcript per million (TPM) > 1. **f** The ability to detect known non-poly(A) genes was comparable across the three methods tested. Each row represents a non-poly(A) gene. Each column represents a sample derived from the indicated method. Gene expression levels [log10(TPM + 1)] are shown using the color scale in the figure. Source data are provided as a Source Data file.

minimal batch effects in our experiments (Supplementary Fig. 4e, f; see scRR3 batches 1 and 2), consistent with our previous report on scRamDA-seq[14]. Nonetheless, scRR-seq protocols showed high coverage across the entire transcript length, including long transcripts, similar to scRamDA-seq (Fig. 1c, d).

Our scRR-seq-RNA detected ~80% of genes and transcripts that were detected by the original scRamDA-seq (Fig. 1e and Supplementary Fig. 4g). scRR-seq-RNA also detected well-known non-poly(A) RNAs as efficiently as scRamDA-seq (Fig. 1f). The remaining ~20% not detected by scRR-seq-RNA were those that show low expression levels by scRamDA-seq and frequently corresponded to long non-coding (lnc)RNAs (Supplementary Fig. 5a–c). Using published RNA localization data[21], we found that these undetected RNAs tend to be nuclear-localized (Supplementary Fig. 5d, e). This is consistent with the fact that, in scRR-seq-RNA, RNA was collected from the supernatant, which mainly corresponds to the cytoplasmic fraction. We further confirmed that a subset of RNAs, enriched for lncRNAs, remained bound to the beads (Supplementary Fig. 6). These results suggest that RNA abundance and subcellular localization influence detection sensitivity in scRR-seq-RNA. This could potentially explain the decrease in the

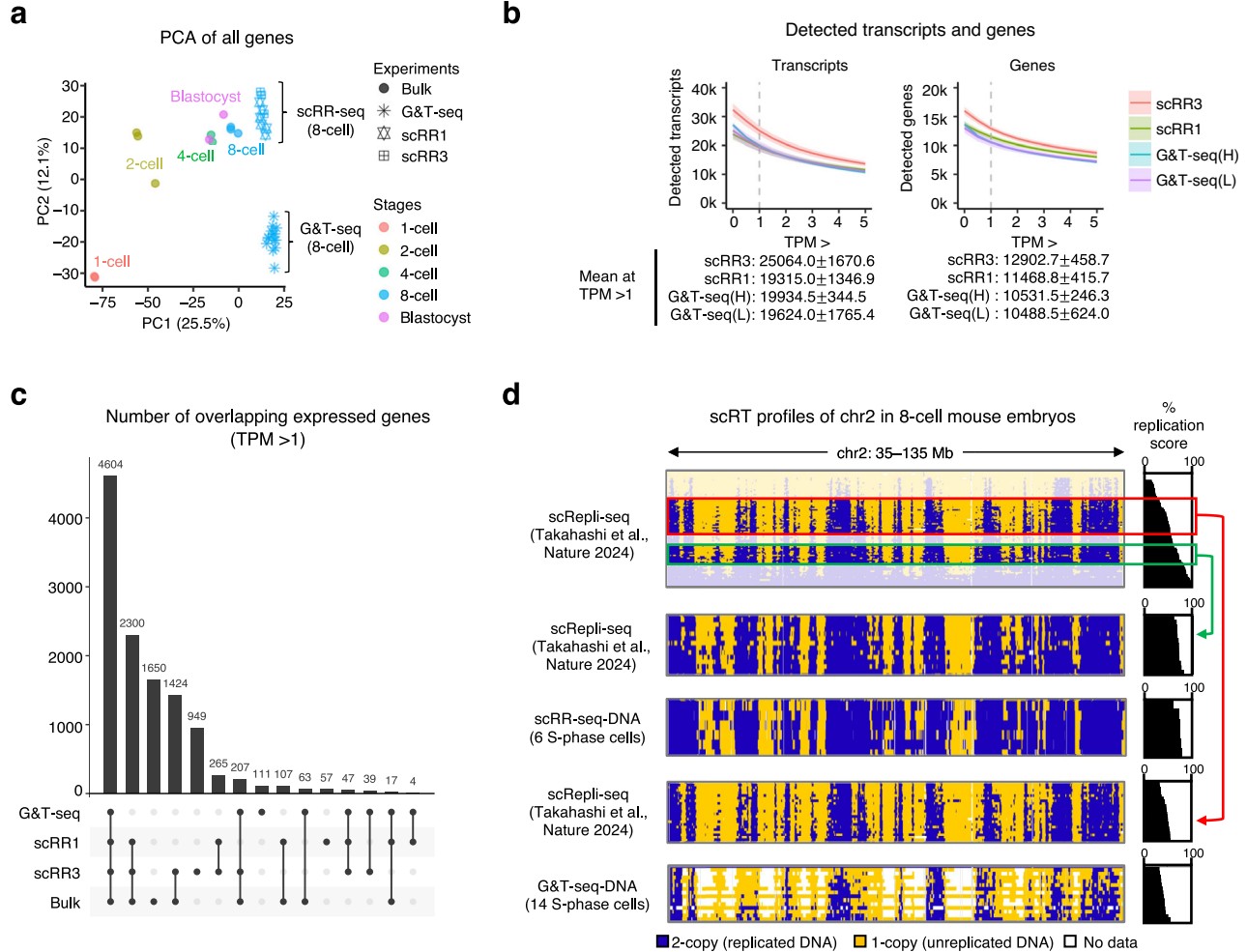

**Fig. 2 | Comparison of scRR-seq and G&T-seq using mouse embryos. a** PCA of all genes. Bulk RNA-seq data of mouse preimplantation embryos[22] were compared with scRR-seq and G&T-seq data[4]. **b** Detection rates of transcripts and genes expressed in 8-cell embryos across different expression level thresholds for each method. The values at the bottom show the mean number of transcripts or genes with SDs at TPM > 1. G&T-seq data were divided into two groups based on initial read counts. G&T-seq(H) includes seven samples with read counts comparable to or higher than those from scRR-seq-DNA, while G&T-seq(L) includes seven samples with lower initial read counts than scRR-seq-DNA. The line and color-shaded areas represent means and SDs, respectively. **c** Number of overlapping genes expressed in 8-cell embryos among the four methods shown (TPM > 1). **d** Comparison of scRT profiles of 8-cell embryos derived from different methods. The top panel shows whole-S scRT profiles of 8-cell embryos obtained via scRepli-seq[16]. A 100-Mb region of chromosome 2 (chr2) is shown. Green and red boxes indicate the subsets of cells used for comparison with scRR-seq-DNA and G&T-seq-DNA[4], respectively. Yellow and blue represent unreplicated and replicated bins, respectively; white represents bins with no data. Source data are provided as a Source Data file.

number of genes detected compared to scRamDA-seq. Overall, however, scRR-seq-RNA data were comparable to scRamDA-seq data obtained individually.

### scRR-seq outperforms G&T-seq

We next asked how scRR-seq compared with G&T-seq[4], a representative single-cell DNA/RNA-seq method that integrates PicoPlex for WGA and DNA analysis and Smart-seq2 for RNA profiling. To this end, we generated scRR-seq (scRR1 and scRR3) data from 8-cell stage mouse embryos and compared them with publicly available 8-cell embryo G&T-seq data (Supplementary Data 1). First, we focused on the RNA-seq component. We adjusted the NGS reads of our scRR-seq-RNA to ~3 M reads/sample to match the G&T-seq RNA-seq data (G&T-seq-RNA hereafter). Then, we evaluated these scRNA-seq data by comparison with bulk RNA-seq data from preimplantation embryos[22]. Using all genes for the analysis, 8-cell scRR-seq-RNA samples clustered more closely with bulk RNA-seq data of 8-cell embryos than 8-cell G&T-seq-RNA samples (Fig. 2a and Supplementary Fig. 7a).

While the overall mapping rates were higher for G&T-seq-RNA (Supplementary Fig. 7b), scRR-seq-RNA, particularly scRR3, detected ~10–20% more transcripts and genes with transcripts per million (TPM) >1 (Fig. 2b), suggesting that scRR-seq-RNA can capture more alternative splicing or isoforms than G&T-seq-RNA. This is probably because scRR-seq-RNA theoretically captures all total RNA, including unspliced forms, while G&T-seq-RNA captures only poly(A) RNAs (Supplementary Fig. 7c). Expressed genes (TPM > 1) in scRR-seq-RNA overlapped more with bulk RNA-seq data than G&T-seq-RNA (Fig. 2c). In addition, the expression levels of well-known non-poly(A) transcripts and histone genes were also higher in scRR-seq-RNA than G&T-seq-RNA and bulk RNA-seq data (Supplementary Fig. 7d). When analyzing embryos, we found that scRR1 showed a bias towards the 3′ end of transcripts (Supplementary Fig. 7e) and detected fewer transcripts than scRR3 (Fig. 2b), which was not the case with cultured single cells collected by FACS (Supplementary Note 1). Nonetheless, scRR-seq overall showed superior sensitivity over G&T-seq for scRNA-seq analysis.

Next, we compared scRR-seq-DNA with DNA-seq data from G&T-seq (G&T-seq-DNA hereafter) for their DNA detection capacity and sensitivity. Again, we adjusted the NGS reads of our scRR-seq-DNA to match those of the G&T-seq-DNA (~ 4 M reads/sample). We found that scRR-seq-DNA exhibited a ~ 50% mapping rate to the genome, while G&T-seq-DNA showed only ~10% (Supplementary Fig. 7f; after removing duplicates). As a result, the genome coverage of scRR-seq-DNA was ~3.2 times higher than that of G&T-seq-DNA (5.1% ± 0.6% and 1.6% ± 0.6%, respectively (mean ± SD); Supplementary Data 1). The lower WGA efficiency observed in G&T-seq may result from DNA loss during the DNA/RNA separation procedure, which involves more steps than scRR-seq.

We also observed significantly higher median-absolute-deviation (MAD) scores for G&T-seq-DNA (Supplementary Fig. 7g). The MAD scores are routinely used to evaluate the uniformity of DNA-seq data[13], where high MAD scores suggest a high level of noise in the data distribution. These observations make scRR-seq a better option for scDNA-seq and scRT analysis.

We obtained G1-like and G2/M-like cells from 8-cell embryos (Supplementary Fig. 8a), which were used to correct the mappability of S-phase cells (Supplementary Fig. 8b). We successfully generated scRT profiles of 8-cell embryos from scRR-seq-DNA at 40-kb resolution, which was challenging for G&T-seq (Fig. 2d), possibly due to its low coverage (Supplementary Fig. 7f and Supplementary Data 1). Taken together, scRR-seq showed enhanced resolution and sensitivity compared to G&T-seq for both DNA-seq and RNA-seq, especially the former.

## scRR-seq captures gene expression dynamics and markers during S-phase progression

As scRT data allows one to tell the S-phase timepoint of each cell (Supplementary Fig. 9), we reasoned that scRR-seq data should reveal gene expression dynamics during S-phase progression at an unprecedented temporal resolution and may help identify S-phase progression markers, which have never been reported before. Therefore, we generated scRR-seq data from hTERT-RPE1 cells and CBMS1 mESCs throughout the S-phase. First, we used scRR-seq-DNA to generate a whole-S RT profile, which showed that we successfully captured the entire S-phase of hTERT-RPE1 cells (Fig. 3a) and CBMS1 mESCs (Fig. 3b and Supplementary Fig. 10). The whole-S RT profile of CBMS1 mESCs closely resembled the profile obtained previously by scRepli-seq[13] (Supplementary Fig. 11a). Then, we used our hTERT-RPE1 and CBMS1 mESC scRR-seq-RNA data to find S-phase progression marker candidates. We sorted cells based on their percentage replication scores and searched for genes that showed significant gene expression changes during S-phase progression. Using the generalized additive model (GAM) fitting, we identified 52 significant dynamic genes (FDR < 0.05) in hTERT-RPE1 and 55 genes in CBMS1 mESCs (FDR < 0.01) and divided them into three groups based on their expression patterns (Supplementary Fig. 11b, c and Supplementary Data 2). Gene Ontology (GO) analysis revealed that these identified genes were significantly overrepresented in processes related to cell division (Supplementary Data 2). Only a few genes were shared between the two cell lines (Supplementary Data 2).

Utilizing these identified genes for diffusion map analysis (DMA), we observed a clear trajectory of S-phase progression (Fig. 3c). However, these human S-phase progression markers were unable to reveal the S-phase progression trajectory in mice and vice versa, suggesting the species- or cell-type-specificity of markers (Fig. 3c and Supplementary Fig. 12a–e). Lastly, we applied these identified S-phase progression markers of CBMS1 to another mESC scRNA-seq dataset[14] to assess their reproducibility. We obtained a trajectory of S-phase progression relatively similar to that of CBMS1 mESCs (Fig. 3d), confirming the robustness of these markers. Taken together, our results demonstrate that scRR-seq enables the comprehensive analysis of gene expression dynamics during S-phase progression at an unprecedented temporal resolution. Moreover, it also allows the identification of a set of S-phase progression markers for a given cell type and species.

## scRR-seq reveals a limitation of existing cell-cycle stage assignment

Cell-cycle markers are commonly used to assign cell-cycle stages to scRNA-seq datasets. For example, Seurat[23], a widely used scRNA-seq analysis pipeline, employs a predefined set of S and G2/M markers from human cells[24] to classify cells into specific cell-cycle stages. Since our scRR-seq datasets capture both the S-phase state and gene expression, we leveraged this dual capability to assess the accuracy of the existing cell-cycle markers and cell-cycle stage assignments.

We used Seurat and its default cell-cycle marker sets to assign cell-cycle stages to individual cells in our whole-S-derived hTERT-RPE1 and CBMS1 mESCs scRR-seq datasets, based on the scRR-seq-RNA data. We then compared these assignments with the percentage replication scores derived from the corresponding scRR-seq-DNA data.

Unexpectedly, while all cells were in S-phase based on the percentage replication scores, Seurat assigned G1, S, and G2/M phases to these cells (Fig. 3e, f). In hTERT-RPE1 cells, while the S and G2/M phase assignments were relatively reasonable and marked cells with low and high percentage replication scores, respectively, some cells were incorrectly labeled as G1 (Fig. 3e).

In contrast, in CBMS1 mESCs, the majority of the cells were assigned as G1 (Fig. 3f). The proportion of cells assigned as S and G2/M was noticeably low, which corresponded to those with low (<40%) and high (>70%) replication scores, respectively (Fig. 3f). Given that Seurat's default markers are derived from human tumor cells[24], our findings suggest that cell-cycle markers may be cell-type or species-specific (Supplementary Fig. 12f). Overall, these results underscored a limitation of the current markers or methodology for cell-cycle stage assignment, particularly when applied across different cell types or species.

## scRR-seq allows haplotype/allele-specific RT and expression analysis

To explore the feasibility of haplotype/allele-specific analysis, we utilized our in-house single-nucleotide polymorphisms (SNPs) information[25] and analyzed scRR-seq data from hTERT-RPE1 cells, focusing on the active and inactive X chromosomes (Fig. 4a, Supplementary Fig. 13a–c). While haplotype-specific scRT profiles of autosomes were consistent with population-based haplotype-specific BrdU-IP Repli-seq (a conventional RT profiling method that utilizes immunoprecipitation of BrdU-substituted DNA[13]) (Supplementary Fig. 13a), haplotype-specific scRT profiles of the X chromosomes (chrX) showed a scattered pattern (Fig. 4a), possibly due to the low SNP density on chrX (Supplementary Fig. 13b). Despite this variability, we could still observe asynchronous RT of certain loci on chrX, with the haplotype-a being replicated later than haplotype-b (Fig. 4a, magnified red boxes). Given that late RT is a defining feature of the inactive X chromosome (Xi), our data suggest that haplotype-a chrX corresponds to the Xi. These results also suggest that the human Xi's RT may be less uniformly late compared to the mouse Xi[25] (Supplementary Note 2 and Supplementary Fig. 14).

To validate this chrX observation genome-wide, we focused on the informative genes [defined as genes with SNP-containing reads (both haplotype-a and b) > 6 in more than 50% of cells in a given scRR-seq-RNA data] and analyzed the gene expression ratio of haplotype-a in hTERT-RPE1 [=haplotype a/(a + b)]. We identified 1878 informative genes using whole-S scRR-seq-RNA data (see "Methods", Supplementary Fig. 13c and Supplementary Data 3). Among the 38 monoallelic genes identified, 14 were found to be X-linked genes, which exhibited a low expression ratio of haplotype-a close to zero, except for those previously identified as escapees (Supplementary Data 3; seven

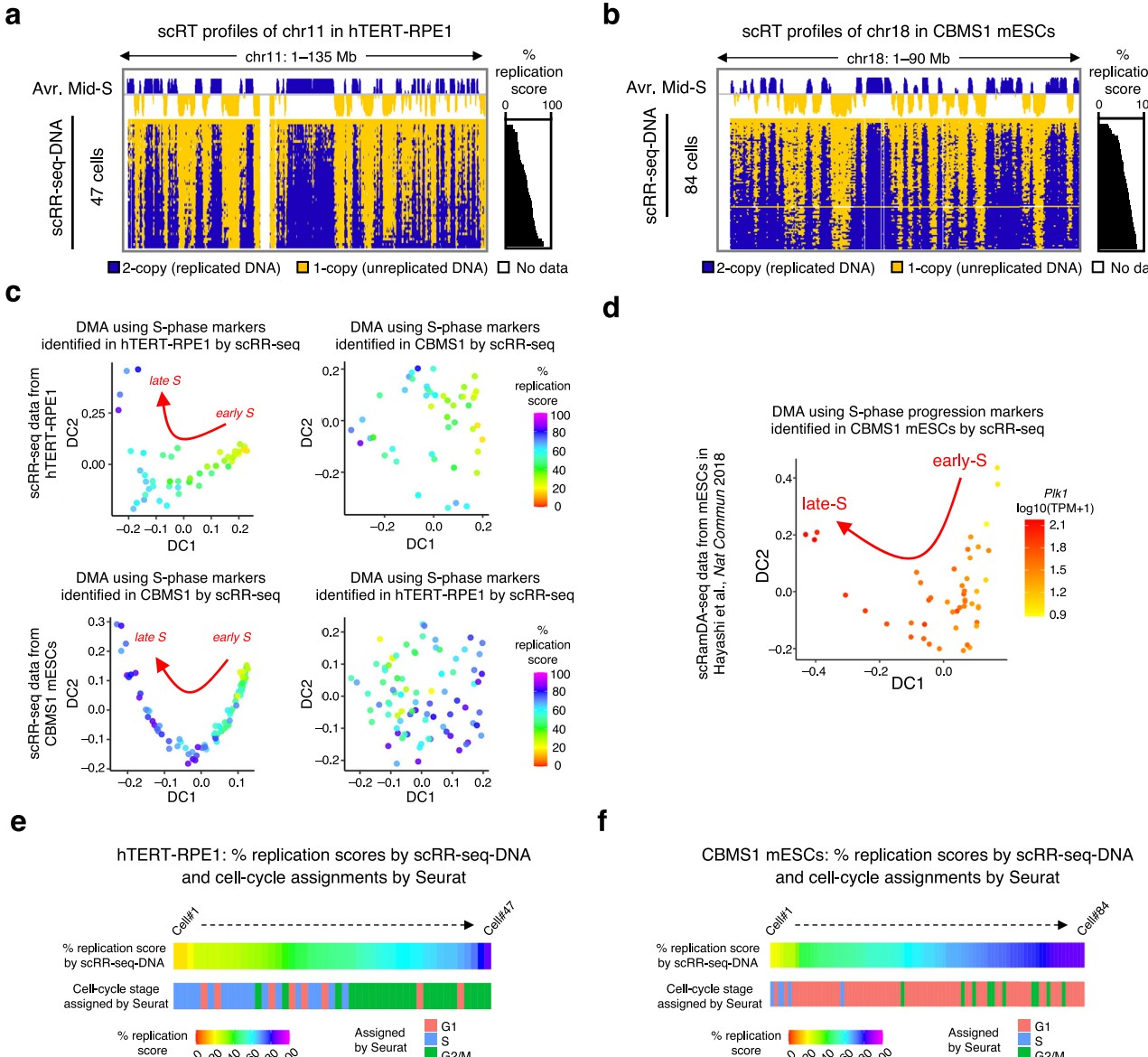

**Fig. 3 | scRR-seq identifies S-phase progression markers. a** Whole-S scRT profiles of hTERT-RPE1 cells derived from scRR-seq-DNA. Chromosome 11 (chr11) is shown. Average RT profiles of mid-S cells (40–70% replication scores) are shown at the top (Avr. Mid-S). **b** Whole-S scRT profiles of CBMS1 mESCs derived from scRR-seq-DNA. Chr18 is shown. Average RT profiles of mid-S cells (40–70% replication scores) are shown at the top (Avr. Mid-S). As the scRR1 and scRR3 methods produced similar benchmarks for CBMS1 mESCs (Supplementary Fig. 10), we combined all scRR-seq-DNA data and generated a whole-S RT profile. **c** Diffusion map analysis (DMA) of hTERT-RPE1 and CBMS1 mESCs using identified S-phase progression markers from

scRR-seq, as indicated. Each cell is color-coded by its percentage replication score. **d** DMA of mESC scRamDA-seq data[14] using the S-phase progression markers identified by scRR-seq-RNA in CBMS1 mESCs. *Plk1* was used as a reference gene, as its expression increases during cell-cycle progression. Each cell is color-coded by expression level of *Plk1*. **e, f** Comparison between percentage replication scores from scRR-seq-DNA and cell-cycle phase assignments from Seurat using its default cell-cycle markers, in hTERT-RPE1 cells (**e**) and CBMS1 mESCs (**f**). Each cell is color-coded by its percentage replication score (top) and Seurat-assigned cell-cycle phase (bottom). Source data are provided as a Source Data file.

representative X-linked genes are shown in Fig. 4b). This confirms that haplotype-a chrX corresponds to the Xi in hTERT-RPE1, consistent with our scRT data showing late RT of the haplotype-a chrX (Fig. 4a). In addition, several known imprinted genes showed skewed gene expression[26,27]. For instance, *MEST* and *PEG10* expression skewed toward haplotype-a (Fig. 4c). Given that they are both known as paternally imprinted genes and located on chr7, our results reveal that haplotype-a chr7 represents the paternal chr7 in hTERT-RPE1 (Fig. 4c).

We previously reported the strong correlation between allelic RT asynchrony and allelic gene expression imbalance in mESCs (i.e., earlier RT correlates with higher expression)[13]. However, this conclusion is based on a combinatorial analysis of scRepli-seq and population-based

RNA-seq data. We decided to revisit this problem in single cells by allele-specific scRR-seq analysis of CBMS1 mESCs (Fig. 4d, e). Specifically, we examined if asynchronous RT correlates with gene expression imbalance within the same single cell. To this end, we identified 5972 informative genes [defined as genes with SNP-containing reads (both CBA and MSM) > 10 in more than 50% of cells in scRR-seq-RNA data] from scRR-seq-RNA data (see "Methods", Supplementary Fig. 13d and Supplementary Data 3). Monoallelically-expressed genes included *Peg3*, *Peg10*, *Snrpn*, and *Rian*, which are known mouse imprinted genes[28,29]. Other known imprinted genes, such as *Igf2r* and *Slc38a4*, showed biallelic expression (Supplementary Fig. 13e and Supplementary Data 3).

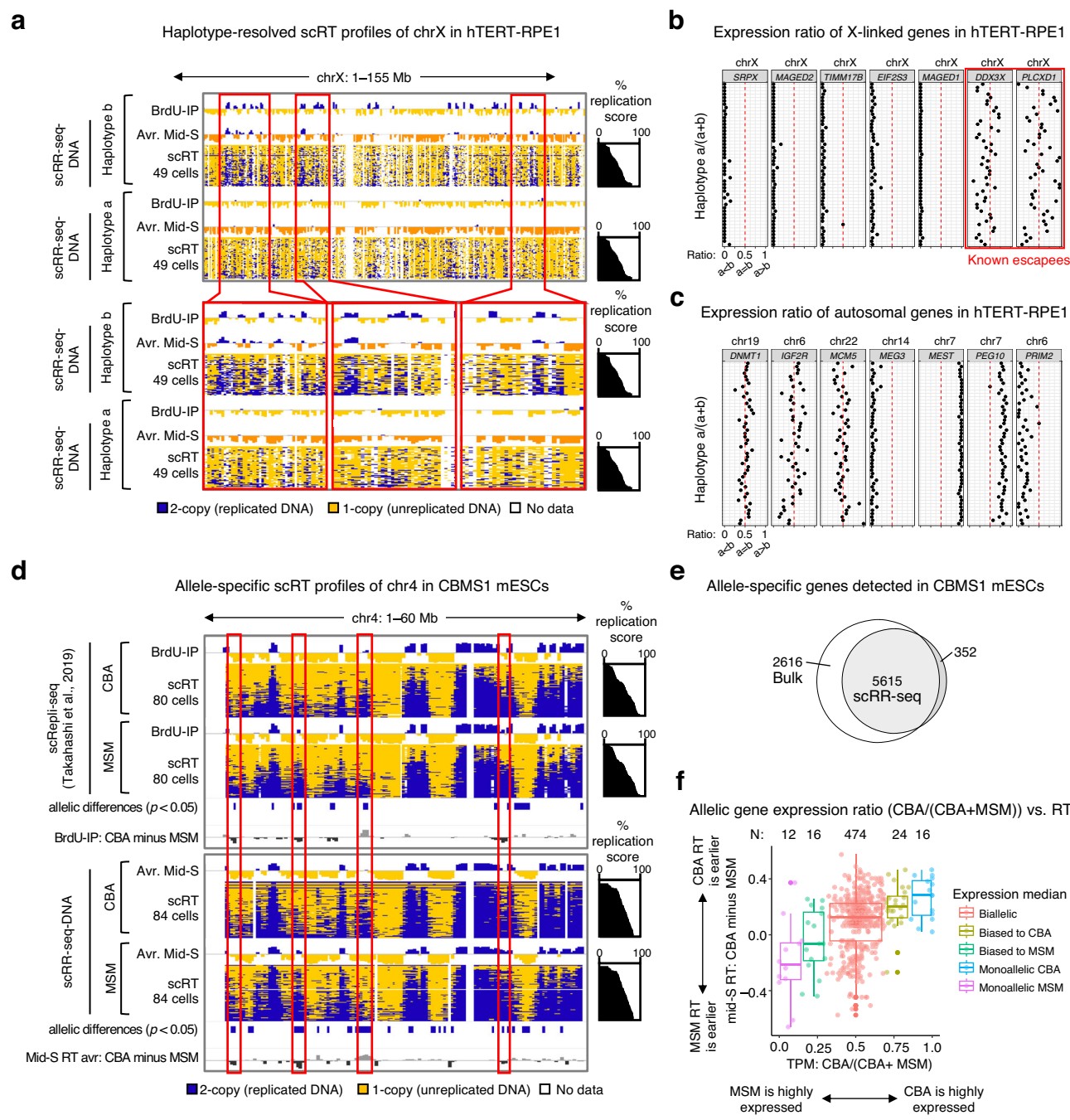

**Fig. 4 | Haplotype/allele-specific analysis by scRR-seq. a** Haplotype-specific scRT profiles of chromosome X (chrX) in hTERT-RPE1 derived from scRR-seq-DNA at 400-kb resolution. For comparison, haplotype-specific BrdU-IP RT profiles are also shown[25]. The average mid-S RT profiles (Avr. Mid-S) are derived from cells with 40–70% replication scores. The top panel shows the entire chrX, while the bottom panel provides a zoomed-in view of the regions marked by the red boxes in the top panel. **b, c** Haplotype-specific gene expression ratios [haplotype a/(haplotype a + b)] of representative X-linked (**b**) and autosomal (**c**) genes in hTERT-RPE1 cells derived from scRR-seq-RNA. Each row shows the haplotype gene expression ratio in each cell (black dot). Red boxes indicate known escapees on the human X chromosome. The chromosome for each gene is indicated above the gene name. **d** Allele-specific scRT profiles of chromosome 4 (chr4) in CBMS1 mESCs derived

from scRepli-seq[13] (top) and scRR-seq-DNA (bottom) at 400-kb resolution. Haplotype-specific BrdU-IP RT profiles[13] and average mid-S RT profiles (Avr. Mid-S) are shown for comparison. Red boxes highlight regions with RT differences. **e** Number of informative allele-specific genes detected in bulk RNA-seq[13] and scRR-seq-RNA, showing that scRR-seq-RNA successfully captures the majority of the allele-specific gene expression observed in bulk RNA-seq data. **f** Comparison of allelic gene expression ratios [CBA/(CBA + MSM)] and their corresponding Δaverage mid-S RT values for informative genes in CBMS1 mESCs, using scRR-seq-RNA and scRR-seq-DNA from (**d**). Boxplot whiskers indicate ±1.5× the interquartile range (IQR), with outliers excluded. The box spans the 25th–75th percentiles, and the central line indicates the median. *N* shows number of genes in each group. Source data are provided as a Source Data file.

Interestingly, monoallelic genes and genes with allelic expression imbalance showed a positive correlation with asynchronous RT in CBMS1 mESCs. That is, CBA-allele specific monoallelic genes or genes with biased expression toward the CBA allele exhibited earlier RT on

the CBA loci, and vice versa was true for the MSM loci (Fig. 4f). For instance, *Peg3* and *Snrpn* were primarily expressed from the MSM loci, and consistently, scRT profiles showed earlier replication of the MSM loci for these genes (Supplementary Fig. 13e, f). Likewise, *Rian*

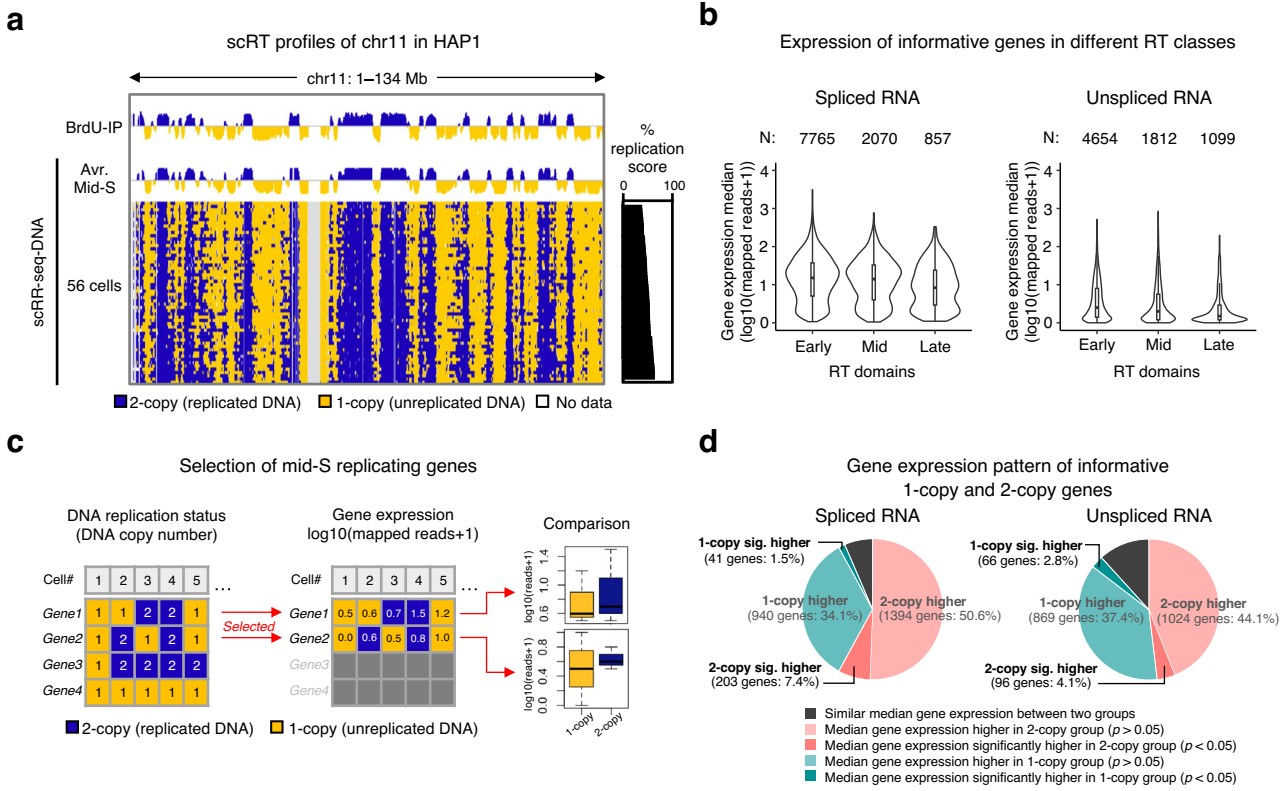

**Fig. 5 | The relationship between HAP1 DNA replication status and gene expression. a** scRT profiles of chr11 in HAP1 cells derived from scRR-seq-DNA. **b** Gene expression levels and RT classes in HAP1. The number of genes in each RT class is indicated. Boxplot whiskers indicate ±1.5× the interquartile range (IQR), with outliers excluded. The box spans the 25th–75th percentiles, and the central line indicates the median. **c** Schematic of how to select genes with variable DNA replication statuses (1- or 2-copy) during mid-S phase. **d** Pie charts showing gene expression patterns of replicated (2-copy) and unreplicated (1-copy) genes during mid-S-phase in HAP1 cells. Source data are provided as a Source Data file.

expression was biased toward the CBA allele, and the CBA *Rian* locus exhibited earlier RT than the MSM counterpart (Supplementary Fig. 13e, f). Overall, these results demonstrate the feasibility of haplotype-specific analysis by scRR-seq, and our data unequivocally demonstrated the positive correlation between transcription and RT within the same single cell.

### scRR-seq reveals a limited link between DNA copy number and gene expression level

Early and late RT correlates with active and inactive transcription, respectively[30]. However, it remains unclear whether the increase in DNA copy number during S-phase plays any role in this relationship. In prokaryotes, transcription levels tend to increase as DNA is replicated (i.e., upon copy number increase), whereas in eukaryotic cells, transcription levels remain relatively static during S-phase[31]. However, these observations were made in organisms with small genome sizes or for a few representative genes by imaging or population-based data[32–34], which were low in resolution and might not reflect reality. To revisit this problem, we leveraged scRR-seq to assess the relationship between DNA replication state and gene expression genome-wide at the single-cell level. The human haploid HAP1 cell line was used to avoid any issues arising from differences between homologous chromosomes.

First, we evaluated the relationship between gene expression and RT. To do so, we used scRT profiles and calculated an average mid-S RT profile of HAP1 cells at 40-kb resolution, which resembled the population-based RT profiles from the BrdU-IP experiment (Fig. 5a). We categorized RT domains into three classes based on the average mid-S RT profile. The range from minimum to maximum was equally

divided into nine fractions. RT domains with average mid-S RT values higher than the 7th fraction (>0.213), in between the 3rd and 7th fractions, and less than the 3rd fraction (<−0.287) were defined as early, mid, and late-S RT domains, respectively. We analyzed both spliced and unspliced RNA expression based on the reasoning that the latter might better reflect the nascent transcript level. As expected, genes located in early-S RT domains showed higher expression levels compared to those in mid and late-S RT domains (Fig. 5b).

To examine if replication status influences gene expression, 40-kb binarized scRR-seq-DNA data were used to assign a DNA copy number to every gene in a given cell (Fig. 5c). We then selected 1-copy (unreplicated) and 2-copy (replicated) genes during mid-S phase (40–70% replication score) for downstream analysis. Genes consistently showing 1-copy or 2-copy status for more than 80% of mid-S cells were excluded, and we analyzed both spliced and unspliced RNA. Only genes that showed mapped reads > 0 in more than 50% of all cells were selected as informative genes, which were then separated into two groups based on their DNA copy number. About 58% of spliced and 48% of unspliced RNAs showed higher median expression in the 2-copy gene group, while 36% and 40%, respectively, were higher in the 1-copy group. However, most of these differences were not statistically significant (Fig. 5d and Supplementary Data 4). Only 8.9% and 6.9% of all spliced and unspliced RNAs, respectively, showed significant differences in gene expression levels between the two groups. Consistent with previous studies, our results suggest that changes in DNA copy number during S-phase have a limited impact on gene expression. It is more likely that gene expression is associated with RT and its related chromatin features than copy number changes during S-phase in mammalian cells.

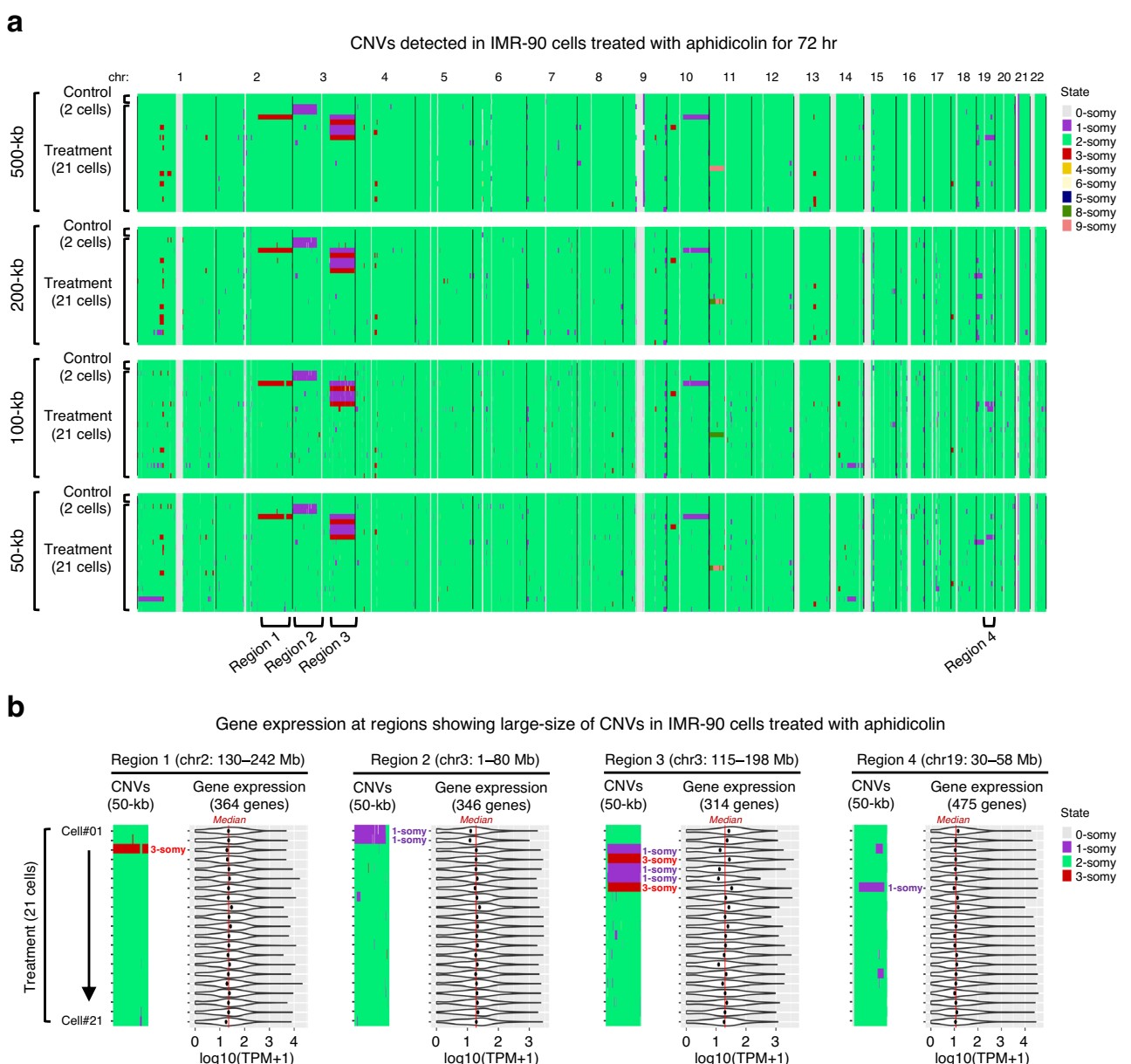

**Fig. 6 | CNVs in IMR-90 cells treated with low-dose aphidicolin. a** Whole genome CNV plots of human IMR-90 cells treated with 0.4 μM aphidicolin for 72 h, compared to untreated control cells shown at the top of each panel. CNVs were analyzed at four bin-size settings: 500 kb, 200 kb, 100 kb, and 50 kb. **b** Regions 1, 2, 3, and 4 indicated in (**a**) are representative CNV-affected regions in IMR-90 cells treated with aphidicolin. Each row shows zoomed-in CNV profiles based on 50-kb CNV analysis (CNV panel) and corresponding gene expression levels (gene expression panel) for individual cells. Violin plots display the expression levels of genes [log10(TPM + 1)] located within each region, with the number of analyzed genes indicated at the top. Only genes with TPM > 1 in more than 50% of cells are shown. Black dots represent the median gene expression of genes in each region per cell, while red lines indicate the median gene expression across all cells for the same genes.

## scRR-seq can detect CNVs induced by DNA replication stress

To assess the CNV detection sensitivity of scRR-seq, we induced chromosome breaks in a diploid female human fibroblast cell line IMR-90 by exposing them to low-dose aphidicolin, a well-known DNA polymerase alpha inhibitor commonly used to induce DNA replication stress. After 72 h of 0.4 μM aphidicolin treatment, G1 cells were collected and subjected to scRR-seq. As expected, we detected various CNV patterns in aphidicolin-treated but not untreated control cells. Large chromosomal abnormalities (>20 Mb)[35] were reliably detected with a resolution of 50 to 500 kb using scRR-seq-DNA, while smaller abnormalities exhibited variability depending on the genomic bin size setting (Fig. 6a). Our data also demonstrates the potential of scRR-seq to detect CNVs smaller than 1000 kb (Supplementary Fig. 15a; at 50 and 100-kb bins).

Next, we examined the relationship between CNVs and gene expression. Because aphidicolin inhibits DNA replication and cell-cycle progression, aphidicolin-treated cells naturally exhibited different expression profiles compared to untreated cells (Supplementary Fig. 15b). Therefore, we instead compared gene expression among aphidicolin-treated cells with and without chromosomal abnormalities. We found that while some chromosomal regions with CNVs showed gene expression alteration, others did not (Fig. 6b). For example, a cell with a large 3-copy CNV on chromosome 2 and a cell with a large 1-copy CNV on chromosome 19 showed median gene expression levels comparable to other cells (Fig. 6b, regions 1 and 4). In contrast, cells with a 1-copy CNV on chromosome 3 exhibited reduced median gene expression, whereas cells with a 3-copy CNV in the same region showed increased gene expression (Fig. 6b, see regions 2 and 3). This suggests a

complex nature of gene regulation in cells with chromosomal abnormalities and implies that relying solely on RNA-seq is insufficient to determine chromosomal abnormalities. In summary, scRR-seq allows one to not only detect CNVs but also assess the impact of CNV on gene expression in the same cell.

## Discussion

In this study, we established a single-cell multi-omics technology that enables high-resolution DNA replication profiling and full-length total RNA sequencing from the same single cell by combining our home-made single-cell technologies, scRepli-seq and scRamDA-seq. scRR-seq can generate DNA-seq and RNA-seq data that are highly comparable to those generated individually by the original methods (Fig. 1) and out-performs other single-cell DNA/RNA-seq methods such as G&T-seq (Fig. 2). scRR-seq also allows haplotype-specific analysis genome-wide at the single-cell level (Fig. 4). In addition, the integration of genome-wide replication and transcription data obtained from scRR-seq led to a much better understanding of the cell-cycle dynamics of gene expression and the identification of S-phase progression markers in human cells and mESCs (Fig. 3). Furthermore, scRR-seq allowed us to explore the precise relationship between DNA replication and transcription genome-wide in individual human cells (Fig. 5). In conclusion, scRR-seq is a powerful methodology that can provide an unprecedented, comprehensive view of DNA replication and gene expression within the same single cell.

scRR-seq-DNA can generate a single-cell genome-wide DNA copy number profile. From our benchmarking, scRR-seq-DNA was nearly identical to scRepli-seq performed individually, generating high-resolution DNA-seq data and scRT profiles from single S-phase cells (up to 40 kb in resolution). Since RT correlates well with A/B compartments[30,36,37], we can leverage this correlation to predict the A/B compartment profiles, which is challenging with the current single-cell Hi-C resolution. In addition, our scRR-seq-RNA achieves approximately 70–80% efficiency in detecting transcripts and gene expression compared to scRamDA-seq, which we consider sufficient to obtain valuable transcriptional information from a single cell. Thus, the combination of high-resolution DNA-seq and RNA-seq data from the same cell offers an unprecedented opportunity to explore cellular processes involving transcription, DNA replication, and 3D genome organization.

Although the scDNA-seq component of scRR-seq showed a strong performance comparable to scRepli-seq, the scRNA-seq component is by no means perfect. For instance, despite its ability to detect the majority of RNA molecules, capturing certain lncRNAs and weakly expressed RNAs appears to be a challenge for scRR-seq-RNA, which is at least partly due to the physical loss of nuclear RNA during the DNA/RNA separation step. Nevertheless, scRR-seq-RNA still outperforms other available scDNA/RNA-seq methods in capturing lowly-expressed genes, including non-poly(A) genes, and the number of detected genes (Fig. 2b and Supplementary Fig. 7d). Thus, it is well-suited for generating global expression profiles and identifying cell types.

Using scRR-seq, we could accurately determine the S-phase timepoint of each cell based on the percentage genome replication score and generate snapshot gene expression profiles of individual cells, meaning that we can obtain a moment-by-moment view of gene expression and monitor its dynamic changes during S-phase progression. We used our whole-S scRepli-seq profile derived from scRR-seq to evaluate existing cell-cycle markers commonly used in scRNA-seq analysis. Our results revealed the limitations of these markers (Fig. 3). In addition, we have identified S-phase progression markers in both hTERT-RPE1 and mESCs by scRR-seq. These markers should enable the determination of the S-phase status of any given cell. For instance, when a cell has the same or similar DNA copy number calls among all genomic bins throughout the genome (either 1-copy or 2-copy), the hidden Markov model (HMM) analysis used in our pipeline fails to

binarize scRepli-seq data, resulting in difficulty distinguishing whether the cell is in the very early-S or very late-S phase. This issue should be solved by incorporating S-phase progression marker information.

Since scRR-seq enables the simultaneous generation of genome-wide DNA replication and gene expression data at the single-cell level, we revisited the effect of DNA replication state on gene expression, which tends to increase as DNA is replicated in prokaryotes[31]. We applied scRR-seq to human haploid HAP1 cells and analyzed mid-S replicating genes in mid-S cells because these genes are in a mixture of replicated and unreplicated states in the population (Fig. 5). Consistent with previous studies in human cells that examined subsets of genes[33,34], the majority of genes we analyzed showed similar gene expression levels despite variations in DNA replication status (Fig. 5d), indicating the limited impact of DNA copy number on gene expression during S-phase in mammalian cells. These results reveal a robust nature of transcriptional homeostasis during S-phase progression in mammalian cells.

scRR-seq also enables simultaneous CNV detection and gene expression analysis in non-S-phase cells (Fig. 6). We found that CNVs and gene expression levels are not always positively correlated (Fig. 6b, regions 1 and 4). Since the regulation of gene expression associated with CNVs is complex and not fully understood[38], scRR-seq offers a unique opportunity to explore this relationship at the single-cell level. In particular, scRR-seq could be highly useful in cancer research, where gene expression and CNVs vary between cells and influence tumor progression and therapeutic response[39]. Furthermore, we recently demonstrated that scRepli-seq can identify CNVs in mouse embryos at the single-cell level, regardless of whether cells are in G1 or S-phase[16]. This suggests that scRR-seq could be extended to explore gene expression and CNV associations in S-phase cells, which would be particularly valuable for analyzing rare or limited samples, for instance, single cells derived from tissues in vivo or patient samples.

Regarding the overall cost and time efficiency, we believe scRR-seq is comparable to most existing protocols. It typically requires approximately 2–4 days each for DNA-seq and RNA-seq library preparation, which is similar to standard NGS workflows. The scRR-seq protocol is simple and easy, as it utilizes the commercially available RamDA-seq kit for RNA-seq. This simplifies the workflow and reduces variability associated with manual protocols. In terms of cost, our optimized scRR-seq protocol lowers reagent expenses to about $60 per sample for the combined DNA and RNA-seq library preparations (see Supplementary Data 5). In comparison, most other methods use Smart-seq-based technology as the RNA-seq component, which either requires technically challenging manual protocols or commercial kits that are substantially more expensive (which cost ~$13 and ~$80/sample for RNA-seq alone, respectively)[40]. Although scRR-seq may not provide a major advantage in processing time or total cost, its simplified workflow makes it more user-friendly and easier to adopt in standard laboratory settings.

While scRR-seq is not compatible with high-throughput processing of a large number of cells, it outperforms such methods regarding data resolution and, therefore, is best suited for analyzing cells of interest in depth. With its simplicity, versatility, and broad applicability, scRR-seq should contribute to addressing various biological questions, including those related to DNA replication, genome instability, and transcriptional regulation.

## Methods

### Cell culture

Experiments involving genetically modified organisms were conducted in accordance with the Regulations on the Use of Living Modified Organisms and the regulations of Mie University, and were approved by the Recombinant DNA Experiment Safety Committee of Mie University. Female CBMS1 mESCs[41] were grown in 2i/LIF medium as described[42]. HAP1 cells (Horizon, C859) were cultured in Iscove's

Modified Dulbecco's Medium (SIGMA, # I3390-500ML) containing 10% FBS and 1× penicillin/streptomycin. hTERT-RPE1 cells (Clontech, C4001-1) and IMR-90 cells (JCRB Cell Bank, JCRB9054) were cultured in MEMα with L-Glutamine and Phenol Red (Wako, #135-15175) containing 10% FBS and 1× penicillin/streptomycin.

### Animals

All animal experiments conformed to the Guide for the Care and Use of Laboratory Animals and were approved by the Institutional Committee of Laboratory Animal Experimentation of the RIKEN Center for Biosystems Dynamics Research (protocol number: A2015-06-9). Mouse-related experiments were performed as previously described[16]. C57BL/6 mice, aged 8–12 weeks, were used to produce oocytes and sperm. The temperature, humidity, and light cycle of mouse cages were maintained at 20–24 °C, 45–65% and 12 h–12 h dark–light, respectively. Mouse embryos were generated from in vitro fertilization, cultured, and single cells were collected as previously described[16]. The sex of embryos was not considered in the study design. Single cells from two mouse 8-cell embryos were collected and subjected to scRR-seq.

### Aphidicolin treatment in IMR-90 cells

IMR-90 cells (JCRB Cell Bank, JCRB9054) were incubated for 72 h in MEMα with L-Glutamine and Phenol Red (Wako, #135-15175) containing 10% FBS, 1× penicillin/streptomycin, and 0.4 μM of aphidicolin[43] before collection.

### Sample preparation for scRR-seq

All procedures were carried out at 4 °C where possible. For hTERT-RPE1, CBMS1 mESCs, HAP1, and IMR-90, single cells were sorted into 0.2-ml tubes by a FACS using a Sony SH800 cell sorter (single-cell mode). Cells were stained with 30 μg/ml Hoechst 33342 in 10% FBS/PBS and incubated for 1 h at 37 °C in a temperature-controlled chamber before sorting (Supplementary Fig. 2b). For 8-cell embryos, blastomeres were manually dissociated and single cells were collected into 0.2-ml tubes using a mouth pipette. Subsequently, 1 or 3 μl of complete RamDA-seq lysis buffer (TOYOBO, #RMD-101) containing 2 mg/ml or 0.67 mg/ml Dynabeads™ MyOne™ Carboxylic Acid (Invitrogen, #65011), respectively, was added to each tube. Protocols using 1 μl and 3 μl of complete RamDA-seq lysis buffer were designated as scRR1 and scRR3, respectively. Samples were spun down at $10,000 \times g$ for 1 min at 4 °C, vortexed at 2000 rpm for 1 min at room temperature using ThermoMixer C (Eppendorf, # 5382000023), and spun down again at $1000 \times g$ for 5 min at 4 °C. Tubes were then placed on a magnetic stand (MagnaStand v3.2, FastGene, #FG-SSMAG3.2) for 5 min. The supernatant, containing total RNA, was transferred to a new tube and either processed immediately for scRamDA-seq[14] or stored at −80 °C. To the bead-containing fraction, 6 μl of Single Cell Lysis & Fragmentation Buffer (SIGMA, #WGA4-50RXN) containing Proteinase K (SIGMA, # P4850-1ML) was added, followed by centrifugation at $10,000 \times g$ for 1 min at 4 °C. This genomic DNA fraction was either processed immediately for scRepli-seq[13,18] or stored at −30 °C.

### Sample preparation for scRR-seq-RNA

cDNA was synthesized from RNA by using the GenNext®RamDA-seq™ Single Cell Kit (TOYOBO, #RMD-101) according to the manufacturer's instructions, with the following modifications. If previously isolated RNA had been stored, it was first thawed at 4 °C. The RNA was then centrifuged at $10,000 \times g$ for 1 min at 4 °C, heated at 70 °C for 90 s, and held at 4 °C. For genomic DNA digestion, we added 1 μl (for scRR1) or 3 μl (for scRR3) of Genomic DNA Digestion Solution containing ERCC RNA Spike-In [1× RT-RamDA buffer, 1× gDNA remover, 1/500,000 ERCC RNA Spike-In Mix (Thermofisher, 4456740) or 1/25,000 SIRV-Set 4 (Lexogen, 141)]. The mixture was spun down at $10,000 \times g$ for 1 min at 4 °C, vortexed at 2000 rpm for 1 min at 4 °C using ThermoMixer C (Eppendorf), and centrifuged again at $10,000 \times g$ for 1 min at 4 °C. The

sample was then incubated at 30 °C for 5 min and held at 4 °C. Next, we added 1 μl (for scRR1) or 3 μl (for scRR3) of RT-RamDA (RDA) solution [1× RT-RamDA buffer, 1× RT-RamDA Enzyme Mix, 1st not-so-random (NSR) primer for mouse or human]. The sample was again spun down at $10,000 \times g$ for 1 min at 4 °C, vortexed at 2000 rpm for 1 min at 4 °C using ThermoMixer C, and centrifuged at $10,000 \times g$ for 1 min at 4 °C. Reverse transcription was then performed in a thermocycler under the following conditions: 25 °C for 10 min, 30 °C for 10 min, 37 °C for 30 min, 50 °C for 5 min, 98 °C for 5 min, and held at 4 °C. Subsequently, we added 2 μl (scRR1) or 6 μl (scRR3) of Second-strand synthesis solution [1× 2nd strand synthesis buffer, 1× 2nd strand synthesis enzyme, 2nd NSR primer for mouse or human]. The samples were spun down at $10,000 \times g$ for 1 min at 4 °C and vortexed at 2000 rpm for 1 min at 4 °C using ThermoMixer C. For second-strand synthesis (for scRR3), samples were spun down at $10,000 \times g$ for 1 min at 4 °C and incubated in a thermocycler at 16 °C for 60 min, 70 °C for 10 min, and held at 4 °C. For scRR1, samples were spun down at $10,000 \times g$ for 1 min at 4 °C and incubated in a thermocycler at 16 °C for 60 min, 80 °C for 15 min, and held at 4 °C. All samples were finally centrifuged at $10,000 \times g$ for 1 min at 4 °C and either processed immediately for library preparation or stored at −80 °C.

### Library preparation for scRR-seq-RNA

scRamDA-seq NGS libraries were prepared using the Nextera® XT Library Prep Kit 24 Samples (Illumina, #15032350, #15032352) and Nextera® XT Index Kit v2 Set A 96 indexes-samples (Illumina, #15052163) according to the manufacturer's instructions with the following modifications.

For RNA samples from the scRR3 protocol (total volume: 15 μl), we added 27 μl of 1/4× Agencourt AMPure XP beads (Beckman Coulter, A63881), centrifuged at $1000 \times g$ for 1 min, and vortexed at 2000 rpm for 2 min at room temperature using ThermoMixer C. Samples were incubated at 25 °C for 5 min in ThermoMixer C, spun down again at $1000 \times g$ for 1 min, and placed on a magnetic stand for 2–5 min. The supernatant was discarded, and beads were washed twice with 150 μl of 80% ethanol while on the magnetic stand. After drying the beads, we added 3.75 μl of 0.67× Tagmentation DNA buffer, vortexed, and incubated at 25 °C for 5 min in ThermoMixer C. After spinning down at $1000 \times g$ for 1 min, the samples were placed on the magnetic stand for 2–5 min, and the supernatant was transferred to a new tube. Next, we added 1.25 μl of Amplicon Tagment Mix, vortexed at 2000 rpm for 2 min at 4 °C in ThermoMixer C, incubated the samples at 55 °C for 5 min, and held at 10 °C (with 75 °C lid heating function) in a thermocycler. Then, 1.25 μl of Neutralize Tagment Buffer was added, vortexed at 2000 rpm for 2 min at 25 °C in ThermoMixer C, and incubated at 25 °C for 5 min. After centrifugation, we added 3.75 μl of Nextera PCR Master Mix, vortexed at 2000 rpm for 2 min at 4 °C in ThermoMixer C. Next, we added 1.25 μl each of i5 and i7 index adapter and vortexed at 2000 rpm for 2 min at 4 °C in ThermoMixer C. After spinning down, the samples were incubated in a thermocycler at 72 °C for 3 min, 95 °C for 30 s, 14 cycles of [95 °C for 10 s, 55 °C for 30 s, 72 °C for 30 s], 72 °C for 5 min, and held at 4 °C (with 105 °C lid heating function). We then cleaned up the library by adding 15 μl of AMPure XP beads to the solution, vortexed at 2000 rpm for 2 min at 25 °C in ThermoMixer C, and incubated the samples at 25 °C for 5 min. After spinning down at $1000 \times g$ for 1 min, tubes were placed on a magnetic stand for 5 min, the supernatant was discarded, and the beads were washed twice with 150 μl of 80% ethanol while placed on the magnetic stand. After drying, we added 24 μl of TE (pH 8.0) to the beads and vortexed the samples. Samples were incubated at 25 °C for 2 min in ThermoMixer C, then spun down at $1000 \times g$ for 1 min before being placed on the magnetic stand for 2–5 min. The supernatant was transferred to a new tube, and the library concentration was quantified by Qubit (Thermo Fisher Scientific). Sample DNA sizes were quantified by Agilent TapeStation with High Sensitivity D1000 ScreenTape. Optimal libraries showed

sample concentrations of >1 ng/μl and fragment sizes ranging from 100–600 bp.

For RNA samples from the scRR1 protocol (total volume: 5 μl), we took 1 μl of sample solution and added 2.2 μl of Tagmentation DNA buffer, spun down at $1000 \times g$ for 1 min at 4 °C, and vortexed them at 2000 rpm for 2 min at 4 °C in ThermoMixer C. After another spin-down, 1 μl of Amplicon Tagment Mix was added and vortexed again at 2000 rpm for 2 min at 4 °C in ThermoMixer C. We incubated the samples at 55 °C for 10 min and held them at 10 °C (with 75 °C lid heating function) in a thermocycler. Next, we added 1 μl of Neutralize Tagment Buffer, vortexed at 2000 rpm for 2 min at 25 °C in Thermo-Mixer C, and incubated the samples at 25 °C for 5 min. After spinning down, we added 3 μl of Nextera PCR Master Mix, vortexed at 2000 rpm for 2 min at 4 °C in ThermoMixer C. Next, we added 1 μl each of i5 and i7 index adapters and vortexed the samples at 2000 rpm for 2 min at 4 °C in ThermoMixer C. After spinning down, samples were incubated in thermocycler at 72 °C for 3 min, 95 °C for 30 s, 16 cycles of [95 °C for 10 s, 55 °C for 30 s, 72 °C for 30 s], 72 °C for 5 min, and held at 4 °C (with 105 °C lid heating function). We then cleaned up the library by adding 12 μl of AMPure XP to the solution, vortexed the samples at 2000 rpm for 2 min at 25 °C in ThermoMixer C, and incubated them at 25 °C for 5 min. We spun down the samples at $1000 \times g$ for 1 min before placing the tubes on a magnetic stand for 5 min. The supernatant was dis-carded, and we washed the beads with 100 μl of 80% ethanol twice while the samples were placed on a magnetic stand. After drying the beads, we added 24 μl of TE (pH 8.0) and vortexed the samples. The samples were incubated at 25 °C for 2 min in ThermoMixer C and then spun down at $1000 \times g$ for 1 min before being placed on a magnetic stand for 2–5 min. The supernatant was collected and processed as described above.

### Sample and library preparation for scRamDA-seq

After single cells were sorted into 0.2-ml tubes, we added 3 μl of complete RamDA-seq lysis buffer (TOYOBO, #RMD-101). All sub-sequent steps followed the same procedures as described for the scRR3 protocol in the "Sample preparation for scRR-seq-RNA" and "Library preparation for scRR-seq-RNA" sections.

### Sample and library preparation for scRR-seq-DNA

Sample preparation was performed as described in the scRepli-seq protocol[13,18]. Briefly, 6 μl of genomic DNA (gDNA) solution was sub-jected to WGA using the SeqPlex kit (Sigma, SEQXE) in a 30 μl reaction volume, following the manufacturer's instructions. The amplified gDNA was purified and size-selected using Agencourt AMPure XP SPRI beads (1.7× reaction volume), and the SEQXE adapter sequence was removed by the primer removal enzyme Eco57I (Sigma, SEQXE). The gDNA was further purified with Agencourt AMPure XP SPRI beads (1.8× reaction volume) and eluted in 15 μl of 1/10× Elution Buffer (Qiagen). The DNA fragment size peak should be within 200–600 bp, which was confirmed by Agilent TapeStation with High Sensitivity D1000 ScreenTape. We could easily distinguish single cells from zero or two cells by quantification of DNA with MultiNA, which provided reassur-ance that the gDNA was indeed derived from single cells. Then NGS libraries were constructed using the NGS LTP Library Preparation Kit (KAPA, KK8232) with a SeqCap adapter kit A/B (Roche, 07141530001/ 07141548001) or NEXTflex DNA barcodes (Bio Scientific, NOVA), or the NEBNext® Ultra™ II DNA Library Prep Kit for Illumina® (New England BioLabs, #E7645) with either NEBNext® Multiplex Oligos for Illumina® (Dual Index Primers Set 1, Set 2) (New England BioLabs, #E7600S, #E7780S) or NEBNext® Multiplex Oligos for Illumina® (Index Primers Set 1, Set 2, Set 3, Set 4) (New England BioLabs, #E7355L, E7500L, E7710L, E7730L). Finally, the samples were subjected to NGS on an Illumina HiSeq X Ten or Novaseq X Plus system (150-bp length, paired-end reads).

### Sample preparation for population-based RT profiling by BrdU-IP Repli-seq (HAP1)

We followed the BrdU-IP protocol as described[13]. HAP1 cells were incubated in medium containing 10 mM BrdU (Sigma) for 2 h before cell collection. After trypsinization, the single-cell suspension was fixed in 75% ethanol. For FACS, we stained the fixed cells with propidium iodide (Nacalai, 29037) and used a Sony SH800 cell sorter (ultra-purity mode) to sort early- and late-S-phase cell populations (at least 10,000 cells per fraction). We used a Bioruptor BRII (Sonic Bio) for genomic DNA soni-cation (high-output mode), with ON/OFF pulse times of 30 s/30 s for 12 min in ice-cold water. BrdU-incorporated DNA was immunoprecipi-tated using anti-BrdU antibody (dilution to 12.5 μg/ml by PBS; BD Bios-ciences Pharmingen, 555627). After BrdU-IP, immunoprecipitated DNA samples were subject to WGA with a SeqPlex kit (Sigma, SEQXE). NGS libraries were constructed from early- and late-replicating DNA after WGA using the NEBNext® Ultra™ II DNA Library Prep Kit for Illumina® (New England BioLabs, #E7645), following the manufacturer's instruc-tions, and were subjected to NGS on an HiSeq X Ten system or Novaseq X Plus system (150-bp length, paired-end reads).

### Computation associated with the transcriptome profiling of single cells (scRNA-seq)

All RNA-seq, scRNA-seq, and scRR-seq-RNA datasets were processed using a complete nextflow pipeline, ramdaq v.1.9.2[20]. The following tools were used; Nextflow v.23.04.01, FastQC v.0.11.8, Fastq-mcf v.1.04.807, Hisat v.2.2.0, Samtools v.1.10, Bam2Wig v.3.0.1, Bamtools v.2.5.1, read_distribution v.3.0.1, inter_experiment v.3.0.1, inner_distance v.3.0.1, junction_annotation v.3.0.1, featureCounts v.2.0.1, RSEM v.1.3.1, edgeR v.3.26.5, and MultiQC v.1.9. Human GENCODE v38 and mouse GENCODE vM25 were used as reference genomes. Publicly available bulk RNA-seq data of mouse preimplantation embryos[22] (short-read raw sequencing data; SRA accession: SRP225196), G&T-seq data of 8-cell mouse embryos[4] (ENA accession: PRJEB9051), and scRamDA-seq of mESCs[14] (GEO accession: GSE98664) were used for the analyses.

### Computation associated with the RT profiling of single cells (scRepli-seq)

For non-haplotype/allele-specific analyses, we followed our estab-lished pipeline for scRepli-seq RT analysis as described[13,18]. We mapped scRR-seq-DNA data to the hg38 and mm10 reference genomes for human and mouse cells, respectively, using bwa (v.0.7.17-r1188). Gen-ome coverage of unique mapped reads per sample was computed using genomeCoverageBed from bedtools.

We then analyzed the whole genome scRR-seq-DNA data for each single cell in 40-kb non-overlapping bins (or 80-kb bins where indi-cated) using AneuFinder[44] (v.1.2.1). For binarization, different options were applied to each cell depending on their FACS sorting gates (2-HMM option for G1 FACS gate: most.frequent.state = "1-somy"; 2-HMM option for S-phase FACS gates: most.frequent.state = "2-somy").

To calculate the percentage replication scores, we excluded abnormal chromosomes and the X chromosome from each cell line to avoid bias (chromosomes 10 and X were removed for hTERT-RPE1 analysis; chromosomes 8 and X were removed for CBMS mESC ana-lysis; chromosomes 1, 15, and X were removed for HAP1 analysis). We also filtered out ambiguous regions based on ENCODE Blacklist version 2[45]. Replicated bins (bins with 2-copy state) were then counted and divided by the total number of analyzed bins per cell. scRT profiles were sorted according to their percentage replication scores and visualized using the IGV browser.

For haplotype/allele-specific analyses, we performed haplotype-resolved scRepli-seq as described[18,25], using 400-kb non-overlapping bins. We used our in-house phased hTERT-RPE1 genome based on SNP information aligned to hg19. For mouse cells, we used our in-house

allele-specific CBMS1 genome based on SNP information aligned to mm10.

All average mid-S RT profiles were generated from cells with 40–70% replication scores and computed using our established pipeline[13]. Publicly available non-haplotype and haplotype-specific BrdU-IP profiles and scRT profiles of hTERT-RPE1 and CBMS1 mESCs[13,25], as well as scRT profiles of 8-cell embryos[16], were also used. Published datasets aligned to the mm9 reference genome were converted to mm10 using UCSC LiftOver.

## Quality controls of scRR-seq samples
For scRR-seq-DNA, we first calculated the mapping ratio by dividing the number of reads with MAPQ > 10 by the total number of reads. Cells with a mapping ratio below Q1−1.5 × IQR were excluded. We also calculated the MAD score using the log2 ratio of read counts to the genome-wide median across non-overlapping 200-kb windows. We filtered out cells with MAD scores >0.3 for G1 cells, and <0.4 or >0.8 for mid-S phase cells. After sorting the cells based on their percentage replication scores, we calculated Manhattan distances between samples. Cells with a Manhattan distance greater than Q3 + 1.5 × IQR were excluded. Cells with percentage replication scores below 10% or above 90% were also excluded from downstream analysis.

For scRR-seq-RNA, we first quantified the number of uniquely mapped reads to the genome (using hisat) in each cell and filtered out cells with values below Q1−1.5 × IQR. In addition, we examined uniquely mapped reads to ribosomal RNA (rRNA) and mitochondrial RNA (mtRNA) and filtered out those with values above Q3 + 1.5 × IQR for either. Cells that failed to pass the QC criteria for either scRR-seq-DNA or scRR-seq-RNA were excluded from all subsequent analysis.

## Subcellular localization of RNAs
Cytoplasmic/Nuclear Localization was assessed using the Relative Concentration Index (CN-RCI) obtained from the LncATLAS database (https://lncatlas.crg.eu/). Only genes with available CN-RCI data were included in the comparison.

## Comparison of bulk RNA-seq data of mouse preimplantation embryos with scRR-seq-RNA and G&T-seq
All genes were used for PCA across the bulk RNA-seq, scRR-seq-RNA, and G&T-seq-RNA datasets. Gene-level expression values were log-transformed as (log10(TPM + 1)), and PCA was performed using the prcomp function in R (scale. = TRUE).

## Cell-cycle phase assignment by Seurat
We analyzed CBMS1 mESC scRR-seq-RNA data using the Seurat (v5.0.1) package in R. Gene-level expression data (RSEM) were normalized using the "NormalizeData" function (normalization.method = "LogNormalized", scale.factor = 10,000). The data were then scaled by the "ScaleData" function, with all genes used as features. Next, we utilized the "CellCycleScoring" function (using default S-phase and G2/M-phase genes for s.features and g2m.features, respectively) to predict the cell-cycle phase of CBMS1 mESCs. Subsequently, we ran "RunPCA" using the default S-phase and G2/M-phase marker genes in Seurat as features. PCA plots were color-coded based on cell-cycle phases assigned by CellCycleScoring as described above or by actual percentage replication scores, as indicated.

Alternatively, we ran the "DiffusionMap" function from the destiny package (v.3.12.0) using the same default S-phase and G2/M-phase marker genes as above. Diffusion map plots were similarly color-coded based on the actual percentage replication scores as indicated. We conducted similar analysis for hTERT-RPE1.

## Identification of S-phase progression markers
For CBMS1 mESCs, we used only cells that passed QC for both scRR-seq-DNA and scRR-seq-RNA. We sorted cells according to their percentage replication scores and then searched for genes that showed significant gene expression changes during S-phase progression using an analysis method adapted from Hayashi et al.[14]. We used log-transformed gene-level expression data (derived from RSEM) as a function of the percentage replication score for this analysis. We fitted a GAM to gene-level expression as a smooth function of percentage replication score for each gene using the mgcv R package (version 1.8–16) with the parameter "family = Gaussian(link = identity)." The p-values of the smooth term were assessed using one-sided F-test provided by mgcv, and resulting p-values were adjusted for multiple testing across all genes using the Benjamini–Hochberg procedure to obtain the false discovery rate (FDR). The Akaike information criterion (AIC) was calculated for GAM and an intercept model. Genes with an FDR < 0.01 and an AIC greater for GAM than for the intercept model were called dynamically regulated genes. Next, we selected only expressed genes with a TPM of at least 1 in at least 20% of cells, then used them for hierarchical clustering into 3 clusters using the flash-Clust R package (version 1.01–2) according to Ward's method and 1−Pearson correlation coefficient as the distance. For hTERT-RPE1, we conducted the analysis similarly as described above, but with the FDR threshold set to <0.05.

GO analysis of S-phase progression markers was performed using the GO resource (https://geneontology.org/) with default settings.

## Haplotype-specific gene expression ratio
We used in-house SNPs to generate a diploid genome and transcriptome of hTERT-RPE1 using EMASE[46] (https://github.com/churchill-lab/emase) and the human GRCh37.75 Ensembl reference genome and transcriptome. Because the parental origin of the haplotypes is unknown, we labeled them as haplotype-a and haplotype-b, which do not necessarily correspond to paternal or maternal chromosomes. For example, haplotype-a may represent the paternal allele on chromosome 7, while haplotype-a on chromosome 8 could be of maternal origin. Genes with SNP-containing reads (haplotype a + b) > 6 in more than 50% of cells were defined as informative genes. We calculated the haplotype gene expression ratio [TPM haplotype a/(TPM haplotype a + b)] for each cell and used the median of this ratio to categorize genes into four categories (median ≥0.85 or ≤0.15 defined as "monoallelic", median ≥0.7 but <0.85 defined as "biased to a", median ≥0.15 but ≤0.3 defined as "biased to b", median >0.3 but <0.7 defined as "biallelic"). Imprinted genes were obtained from: https://www.geneimprint.com/site/genes-by-species.Homo+sapiens.

For CBMS1, we similarly used in-house SNPs to generate a diploid genome and transcriptome by EMASE[46] using mouse GRCm38.75 Ensembl reference genome and transcriptome. Genes with SNPs-containing reads (CBA + MSM) > 10 in more than 50% of cells were defined as informative genes. After excluding genes from karyotypically abnormal chromosomes 8 and X, we calculated the allelic gene expression ratio [TPM CBA/(TPM CBA + MSM)] for each cell and used the median of this ratio to categorize genes into four categories (median ≥0.85 or ≤0.15 defined as "monoallelic", median ≥0.7 but <0.85 defined as "biased to CBA", median ≥0.15 but ≤0.3 defined as "biased to MSM", median >0.3 but <0.7 defined as "biallelic"). Imprinted genes were obtained from: https://www.geneimprint.com/site/genes-by-species.Mus+musculus.

To examine the relationship between allelic gene expression and RT in CBMS1 mESCs, we combined allele-specific expression ratios with mid-S phase RT values of the CBA and MSM alleles. First, we selected genes with available allelic mid-S RT data. Informative genes were then classified based on their median allelic expression ratio into five categories (median ≥0.85 defined as "monoallelic to CBA", median ≥0.7 but <0.85 defined as "biased to CBA", median ≥0.15 but ≤0.3 defined as "biased to MSM", median ≤0.15 defined as "monoallelic to MSM", median >0.3 but <0.7 defined as "biallelic"). The RT difference (mid-S RT CBA minus mid-S RT MSM) was plotted against the allelic

expression ratio for each category using a combination of boxplots and individual points to visualize both the distribution and the data points. Individual points represent data from each gene.

### Allelic RT differences between CBA and MSM in CBMS1 mESCs

To detect differences in RT between the CBA and MSM alleles, we performed two-sided Fisher's exact test for each genomic region[13]. Informative RT bins were first selected based on allele-specific binarized RT data, and bins with missing data in more than 20% of samples were excluded. For each bin, a contingency table of replicated and unreplicated reads for each allele was constructed, and a two-sided Fisher's exact test was applied. Bins with $p$-values < 0.05 were considered to exhibit significant allelic differences in RT.

### Analysis of spliced and unspliced RNA counts in relation to DNA replication in HAP1

For a fair comparison, all scRR-seq-RNA data from HAP1 were down-sampled to a library size of 1 million reads per cell. To quantify spliced and unspliced transcripts, aligned BAM files were processed using featureCounts with GENCODE v38 annotations. For spliced RNA, mapped read counts from the featureCounts with exon annotations were used (directly obtained from ramdaq pipeline). For unspliced RNA, we followed the method from Lee et al.[47]. We employed the featureCounts with genebody annotations to obtain genebody-mapped read counts (both exon and intron counts) for each gene. Then, to obtain gene-level intron counts, we subtracted exon-mapped reads from genebody-mapped reads and defined as unspliced RNA counts. Negative values after subtraction were set to zero.

Spliced and unspliced counts for each gene from individual cells were linked to DNA replication status (1- or 2-copy) based on genomic coordinates. For genes spanning multiple binarized RT bins, RT values were averaged across overlapping bins and then binarized as replicated (2-copy) or unreplicated (1-copy).

For downstream comparisons, only genes with counts >0 in at least 20% of samples were retained. Spliced and unspliced RNA counts were then compared between replicated (2-copy) and unreplicated (1-copy) states for each gene. Differences were assessed using a two-sided Wilcoxon rank-sum test. Genes were further summarized into categories based on whether replicated genes (2-copy) show higher or lower spliced and unspliced RNA counts than unreplicated genes (1-copy), with significance thresholds set at $p$-value < 0.05.

### Computation associated with CNV detection in single cells

We mapped G1-phase IMR-90 scRR-seq-DNA data to the hg38 reference genome using bwa (v.0.7.17-r1188). We then analyzed the scRR-seq-DNA data genome-wide for each single cell in 50-kb, 100-kb, 200-kb, and 500-kb non-overlapping bins using AneuFinder[44] (v.1.2.1). We analyzed CNVs using the Aneufinder wrapper with the following parameters: method = "dnacopy", correction.method = "mappability", blacklist = ENCODE Blacklist version 2[45], mappability.reference = merged G1 reads). Genome-wide CNVs were plotted using the "heatmapGenomewide" function in AneuFinder (v.1.2.1) without clustering.

To assess gene expression within CNV-affected regions in single cells, we analyzed G1-phase IMR-90 scRR-seq-RNA data using a complete nextflow pipeline, ramdaq v.1.9.2[20] and human GENCODE v38 as the reference (similar to above). Then, we used gene-level expression data (RSEM) to evaluate gene expression levels within CNV-affected regions in each cell. Cells identified as S-phase contaminants based on scRR-seq-DNA (MAD score > 0.39) and those showing low-quality scRNA-seq-RNA data (low mapping and high duplicate rates) were excluded prior to downstream analysis.

### DNA replication uniformity of the X chromosome

We can assume that all bins on a given chromosome, including the X, exhibit a uniformly unreplicated status in early-S phase cells. As DNA

replication initiates and progresses, this uniformity decreases, reaching its lowest point during mid S-phase, and increases again once replication is completed. To quantify this pattern on the X chromosome, we defined a copy number uniformity score. This score is calculated by comparing the number of unreplicated and replicated bins on the X in each cell. Specifically, the absolute difference between the number of unreplicated and replicated bins is divided by the total number of valid bins. Bins with no data are excluded from the analysis. The score ranges from 0 to 1, where a score of 1 indicates complete uniformity (i.e., all bins are either unreplicated or replicated), and a score of 0 indicates maximum heterogeneity (i.e., equal numbers of unreplicated and replicated bins). Higher scores reflect more uniform replication timing across the chromosome in a given cell, while lower scores indicate greater variability.

### Statistics & reproducibility

**Statistics.** Statistical analyses were performed in R. Statistical parameters, including the statistical tests used are also reported in associated figure legends, Supplementary Data files, main texts, or "Methods" section. Two-tailed unpaired Wilcoxon rank-sum test was used for Fig. 5d and Supplementary Data 4. The statistical methods used for Supplementary Data 2 are described in the "Methods" section, "Identification of S-phase progression markers." Two-tailed Fisher's exact test was used for Fig. 4e and Supplementary Data 3.

**Experimental replicates.** For Fig. 1, mid-S phase RPE1 cells for scRR3 were pooled from two independent biological experiments, whereas scRR1 cells were derived from one biological experiment. For Fig. 2, two mouse 8-cell embryos (biological replicates) were collected and processed in parallel. For Figs. 3a and 4a, whole-S phase RPE1 cells were pooled from two independent biological experiments, each for scRR1 and scRR3. For Figs. 3b and 4d, whole-S phase CBMS1 mESCs were derived from one biological experiment and subjected to scRR1. For Fig. 5a, Mid-S phase HAP1 cells were derived from one biological experiment, and for Fig. 6, IMR-90 cells were derived from one biological experiment. For all single-cell data, multiple single-cell samples were independently processed under identical conditions, and genome-wide datasets were generated to confirm data reproducibility.

**Sample size, randomization, and blinding.** No statistical method was used to predetermine sample size. The sample size was chosen based on previous experience and standards in the field. For whole-S analysis, sample sizes were selected to ensure coverage of cells at various stages throughout S phase[13]. Data exclusion was performed as described in the "Methods" section "QCs of scRR-seq samples," with excluded samples listed in Supplementary Data 1. Randomization is not relevant to this study because no comparisons between experimental groups were made. Blinding was not relevant to this study because all metrics were derived from absolute quantitative methods without human subjectivity. Sample sizes (N) are also indicated in the figures or figure legends.

### Reporting summary

Further information on research design is available in the Nature Portfolio Reporting Summary linked to this article.

## Data availability

The NGS data generated in this study have been deposited in the GEO database under accession code GSE278959. Source data are provided with this paper.

## Code availability

The code used for data analysis in this study is publicly available on GitHub at: https://github.com/mcbmieu/scRR-seq and are publicly

available as of the date of publication. The code is also deposited in Zenodo (https://doi.org/10.5281/zenodo.17138460).

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

## Acknowledgements

We thank J. Elumalai for analyzing the DNA content and percentage replication score of CBMS1 mESCs and mouse neural stem cells; K. Inoue and H. Kiyonari for providing the frozen embryos; A. Matsushima for managing the IT infrastructure. This work was supported by MEXT KAKENHI Grant Numbers 23K23862, 24K21835, 25K02254, and 25H01445, and JST CREST Grant Number JPMJCR20S5 to S.i.T.; RIKEN BDR intramural grants, RIKEN Pioneering Project 'Genome Building from TADs', MEXT KAKENHI Grant Number 20K20582 and 25H00982, and JST CREST Grant Number JPMJCR20S5 to I.H.; JST CREST Grant Numbers JPMJCR21N6 and JPMJCR1926, Projects for Technological Development, Research Center Network for Realization of Regenerative Medicine by Japan (JP21bm0404073), the Japan Agency for Medical Research and Development (AMED) to I.N.; JSPS KAKENHI Grant Numbers 23H02411 to K.N.; RIKEN Incentive Research Project FY2023–2024 to R.P. This research was also supported by the Stage Transition Project of RIKEN BDR to I.H. and I.N. and was partly conducted under the Medical Research Center Initiative for High Depth Omics, Nanken-Kyote and Multilayered Stress Diseases in Science Tokyo.

## Author contributions

R.P. analyzed the majority of the data. T.Y. established the protocols and performed most of the scRR-seq experiments. T.I. (Taito Imada) performed the aphidicolin-induced chromosome breakage experiment. S.T. assisted with the embryo experiments. T.I. (Takako Ichinose), T.H., M.K., M.Y. and H.M. assisted with the scRamDA-seq experiments and analyses. K.N. and C.O. provided SNP data for hTERT-RPE1 and CBMS1 mESCs. S.i.T., I.H. and I.N. conceived and supervised the study and wrote the manuscript with R.P. All authors edited and approved the final manuscript.

## Competing interests

The authors declare no competing interests.
