## [Peer Review file · Nature Communications]

scRepli-RamDA-seq: a multi-omics technology enabling the analysis of gene expression dynamics during S-phase

Corresponding Author: Professor Shin-ichiro Takebayashi

Version 0:

Reviewer comments:

Reviewer #2

(Remarks to the Author)

In this work the authors developed a single-cell multi-omics method by physically separating DNA and RNA using magnetic beads, enabling the integration of scRepli-seq and scRamDA-seq. This new approach achieves comparable performance to the original protocols for both modalities. This method demonstrates several notable strengths. It effectively identifies the cell cycle stages of S-phase cells and discovers new S-phase progression markers, which is particularly significant since the cell cycle is a common confounding factor in scRNA-seq data, and existing annotation methods remain suboptimal. Moreover, the authors provide genome-wide analysis of how changes in gene copy number during replication influence gene expression levels, a capability uniquely enabled by single-cell techniques.

Below are some specific comments:

1. The article provides two protocols: scRR1 and scRR3, mainly differing in lysis buffer volume. Why were these two protocols designed separately, and what application scenarios is each suitable for? In the scRR1 reaction, only 1 μ l of the 5 μ l reverse transcription mixture was used for further reactions. Why is this? Does this step result in RNA loss and lead to experimental dropout?
2. Performing FACS is challenging due to the limited number of cells at the 8-cell stage, and relying solely on HMM analysis makes it difficult to distinguish whether cells are in the G1 or G2/M phase. How was the cell cycle of eight-cell evaluated? Was this assessment based on karyotype or other methods?
3. When concluding that "human Xi's RT is less uniformly late compared to the mouse Xi" (line 278 in the main text and line 1155 in Supplementary Note 2), the authors do not reference any figure or statistical results. Furthermore, no data on the mouse Xi are presented in this study. The basis for this conclusion is unclear.
4. In lines 157-162 on pages 6-7, why are non-poly(A) RNAs detected as efficiently as with scRamDA-seq, but lncRNA detection efficiency is lower? What is the preference of this technique for detecting RNAs? Is the detection efficiency for all such RNAs the same except well-known non-poly(A) RNAs?
5. The discussion on the relationship between DNA copy number and gene expression is highly valuable and effectively showcases the capabilities of the authors' method. I have only one minor question here: as replication progresses, does the overall gene expression level in the cell increase? Could the absolute expression of 2-copy genes increase without being reflected in the $[\log_{10}(\text{TPM}+1)]$ values?
6. It would also be helpful if the authors could discuss the cost and time efficiency of their method compared to other techniques in the Discussion section.
7. Line 260: "We obtained a trajectory of S-phase progression relatively similar to CBMS1 mESCs (Extended Data Fig. 5a)" should instead reference Extended Data Fig. 5d.
8. In Fig. 4d, is the scRT shown for chr4 or chrX? The title and caption indicate chr4, but the genomic range in the figure is labeled as "chrX:1-60."

Reviewer #3

(Remarks to the Author)

In the manuscript, Poonperm et. al. developed the single-cell Repli-RamDA-seq, also called scRR-seq. As a multi-omics co-assay tool, sRR-seq can be applied to profile high-resolution DNA replication and quantify full-length total RNA within the same cell. Specifically, scRR-seq has shown promising performance regarding the DNA-seq and RNA-seq quality when compared to scRepli-seq, scRamDA-seq, and G&T-seq. The authors have also applied scRR-seq to study the dynamics of the cell cycle on transcriptome and found that S-phase-associated genes are cell type-specific, and a uniform cell cycle marker set can not define cell cycle status accurately. Moreover, they have explored the effect of DNA replication on gene expression. Overall, scRR-seq can be a powerful tool for studying cells from the perspective of genome and transcriptome. I also have a few comments:

Major comments:

It is encouraged to release all code for the data analysis of this paper to ensure the result is reproducible. Also, the sequencing data can not be found through GEO.

I noticed that you have cited DR-seq without benchmarking with it. Is there any specific reason for that?

As far as I know, it will largely increase the scope of scRR-seq when you apply it to cancer studies, such as the relationship between CNV, mutations, and gene expression. Will it be able to provide one additional figure for such a case study?

The comparison of identifying S phase cells between the Seurat gene set and SRR-seq is unfair. Will it also be able to generate the data from other known phages as ground truth and build a model to predict cell-cycle status using scRR-seq? The model can be very simple, such as a linear model. Then it is possible to compare the accuracy of the two methods for identifying cell cycle status.

Minor comments:

The comparison between scRR-seq to G&T-seq is not fully fair. For example, in the DNA part, scRR-seq-DNA adjusted reads to about 4M/sample. It should also consider cell number and make sure the similar level of mean read count per cell. This is to address my concern about the potential large difference in cell number per sample between SRR-seq and G&T-seq.

Please also provide additional support for your saying that cell-cycle markers are cell-type-specific. In your provided evidence, the difference may be introduced by different species.

It is not surprising that DNA copy numbers can not influence gene expression in normal cells. As I have mentioned in major comments, the relationship between CNV and gene expression in cancer cells can be more interesting.

Reviewer #4

(Remarks to the Author)

Version 1:

Reviewer comments:

Reviewer #2

(Remarks to the Author)

I would like to thank the authors for their thorough and detailed responses to my comments. The authors have fully addressed all my concerns and suggestions. I recommend this manuscript for acceptance.

Reviewer #3

(Remarks to the Author)

The authors have addressed all the comments.

Reviewer #4

(Remarks to the Author)

Point-by-point response to all of the reviewers' criticisms (reviewer comments in bold italic):

REVIEWER COMMENTS

Reviewer #2 (Remarks to the Author):

In this work the authors developed a single-cell multi-omics method by physically separating DNA and RNA using magnetic beads, enabling the integration of scRepli-seq and scRamDA-seq. This new approach achieves comparable performance to the original protocols for both modalities. This method demonstrates several notable strengths. It effectively identifies the cell cycle stages of S-phase cells and discovers new S-phase progression markers, which is particularly significant since the cell cycle is a common confounding factor in scRNA-seq data, and existing annotation methods remain suboptimal. Moreover, the authors provide genome-wide analysis of how changes in gene copy number during replication influence gene expression levels, a capability uniquely enabled by single-cell techniques.

We would like to thank the reviewer for the positive feedback on our study. We have addressed the reviewer's specific questions and comments in detail below.

Below are some specific comments:

1. The article provides two protocols: scRR1 and scRR3, mainly differing in lysis buffer volume. Why were these two protocols designed separately, and what application scenarios is each suitable for? In the scRR1 reaction, only 1 µl of the 5 µl reverse transcription mixture was used for further reactions. Why is this? Does this step result in RNA loss and lead to experimental dropout?

Thank you for your questions. Below, we address questions separately for clarity:

(1.1) Why were these two protocols designed separately, and what application scenarios is each suitable for?

The scRR3 and scRR1 were developed based on the original RamDA-seq protocol, which includes two recommended lysis volumes: a standard protocol using 3 µl and an optional protocol using 1 µl. Initially, the 3 µl protocol was established and provided robust RNA detection. Then, to enhance the workflow and reduce costs, especially for processing large numbers of samples at once, the 1 µl protocol was established and became the basis for scRR1.

The 1 μ l (scRR1) protocol improves the workflow as follows:

- **Reduced reagent costs:** As reaction volumes are scaled down, the cost is \sim 3 times reduced compared to scRR3 (see Supplementary Table 5).
- **Simplified workflow:** The second-strand synthesis products (cDNA) can be used directly for Nextera XT library preparation without the DNA purification step (Supplementary Fig. 2). This saves hands-on time and reduces the risk of sample loss.
- **Minimized risk of library failure:** In the 3 μ l protocol, DNA purification using AMPure XP is required before library preparation (Supplementary Fig. 2). This includes a washing step with ethanol, followed by elution of DNA (see details in Method, section “Library preparation for scRR-seq-RNA”). However, ethanol contamination is critically problematic. If any ethanol remains in this step, even in small amounts, it can interfere with the library preparation reaction by Nextera XT in the next step and lead to failed or low-quality libraries. Since the 1 μ l protocol skips this purification step, it reduces this risk.

We observed that RNA detection is slightly better in samples prepared using the 3 μ l protocol compared to those prepared with the 1 μ l protocol (Fig. 1e). However, given its advantages in simplicity, cost-efficiency, and throughput, we believe the 1 μ l protocol is a more practical choice overall.

However, it is important to note that RamDA-seq lysis buffer is sensitive to excess carryover volume from cell isolation/sorting, which can negatively affect the downstream cDNA synthesis. While this is generally not an issue when single cells are sorted by flow cytometry, it could cause problems when picking single cells manually, such as when isolating cells from early embryos. For example, in the original 3 μ l protocol, the volume of the cell sample added to the lysis buffer should be less than 0.5 μ l (according to the RamDA-seq manual by TOYOBO). Proportionally, this would require the carryover volume to be less than 0.17 μ l in the 1 μ l protocol, which is technically difficult to achieve when single cells are picked manually. In such cases, it is better to use the 3 μ l protocol instead of the 1 μ l protocol to ensure robust results. Therefore, we provided two protocols, and the choice of protocol should be based on the specific experimental setup and objectives.

In summary, scRR3 is suited for experiments that involve difficult manual cell isolation, and/or when high-quality RNA-seq data is crucial, while scRR1 is best for single-cell sorting by flow cytometry and large-scale experiment setup. For easy comparison of the two protocols at a glance,

we provided a summary table for comparison in Supplementary Note 1. For your convenience, the table is also shown below (Table 1 for Reviewers).

Table 1 for Reviewers: Comparison of scRR1 and scRR3

Feature	scRR1	scRR3
Initial lysis buffer volume	1 μ l	3 μ l
Workflow complexity	Simple (no cDNA purification step required)	More steps (includes cDNA purification)
Hands-on time per library preparation	2–4 days	2–4 days
Cost per sample (DNA and RNA-seq)	Low (~\$60/sample)	High (~\$100/sample)
Compatibility with FACS	High	High
Compatibility with manual cell isolation	Low (requires the carryover volume to be <0.17 μ l)	High (requires the carryover volume to be <0.5 μ l)
Best suited for	Medium- to large-scale studies using FACS	Limited samples and manually isolated samples
Example use case	Cell lines, automation-friendly workflows	Embryo dissection, rare or precious samples
DNA detection sensitivity	High	High
RNA detection sensitivity	Slightly low (~3.5% lower than scRR3 for FACS samples; ~10% lower for manually isolated samples)	High

(1.2) Why is only 1 μ l of the 5 μ l reverse transcription mixture used in scRR1? Does this cause RNA loss or dropout?

In our scRR1 protocol, the Nextera XT library preparation reaction is scaled down to 1/5 \times volume to improve cost-effectiveness for single-cell applications. We found that 1 μ l is the maximum volume of cDNA (reverse transcription mixture) that can be added to this scaled-down reaction without interfering with its reaction efficiency. Importantly, this scaled-down setup does not reduce the success rate of library preparation.

Using 1 μ l out of 5 μ l of cDNA in scRR1 does not cause notable RNA loss or dropout. As shown in Fig. 1e, scRR1 detects only slightly fewer transcripts and genes than scRR3 (less than 5%

difference). This is because the RamDA-seq method amplifies cDNA during reverse transcription using the RT-RamDA approach, which can increase the cDNA amount by more than 10-fold in just 30 minutes (Hayashi et al., *Nat. Commun.* 2018, Fig. 1 for Reviewers). This strong amplification ensures that there is sufficient cDNA material for library preparation, even from a small input. As a result, scRR1 can capture global gene expression at the single-cell level, comparable to the scRR3 protocol.

Fig. 1 for Reviewers. Figure 1b from Hayashi et al., *Nat. Commun.* 2018 shows the relative yield of cDNA from mESCs using the RT-RamDA approach, which can increase cDNA yield by more than 10-fold in 30 minutes of reaction.

To improve the manuscript, we updated the experimental scheme and added a description of the library preparation procedures and the rationale behind them in Supplementary Figure 2 as follows:

Supplementary Figure 2: Schematic overview of scRR1 and scRR3 workflows

Supplementary Figure 2: Schematic overview of scRR1 and scRR3 workflows

Detailed protocols are described in the Methods section in the main text. Single cells were collected into 0.2-ml PCR tubes, followed by lysis using complete RamDA-seq lysis buffer (TOYOBO, RMD-101) containing Dynabeads® MyOne™ Carboxylic Acid (beads). If the initial RamDA-seq lysis buffer volume was 3 μ l, we refer to the protocol as scRR3, and if it was 1 μ l, we refer to it as scRR1. Since the RamDA-seq lysis buffer is sensitive to excess carryover during cell isolation/sorting, which can negatively impact downstream cDNA synthesis, it is recommended that the carryover volume of the cell sample added to the lysis buffer be less than 0.5 μ l for scRR3 (as specified in the RamDA-seq manual by TOYOBO). Proportionally, this corresponds to less than 0.17 μ l for scRR1.

The RNA and DNA fractions were separated using a magnetic stand, with genomic DNA being captured by the beads and RNA remaining in the solution. The RNA fraction was then processed for scRamDA-seq using GenNext®RamDA-seq™ Single Cell Kit (TOYOBO). Both protocols are relatively similar. After removing genomic DNA contamination, cDNA was directly amplified from RNA using not-so-random primers and strand displacement amplification (Hayashi et al., *Nat Commun.* 2018).

In the scRR3 protocol, a cDNA purification step using AMPure XP beads is required before library preparation. It is important to completely remove ethanol during this step, as residual ethanol can result in low library yields. However, in the scRR1 protocol, this purification step can be skipped, allowing direct library preparation. For scRR1, only 1 μ l (1/5 \times of cDNA product) is used for library preparation, as this is optimized for the 1/5 \times scaled-down Nextera XT DNA Library Preparation Kit. Despite using a small volume, there is no notable drop in data quality or gene detection. As shown in Fig. 1e, scRR1 detects only slightly fewer transcripts and genes than scRR3 (less than 5% difference). This is because the RamDA-seq method amplifies cDNA during reverse transcription using the RT-RamDA approach, which can increase the cDNA amount by more than 10-fold in just 30 minutes (Hayashi et al., *Nat. Commun.* 2018). This strong amplification ensures that there is sufficient cDNA material for library preparation.

For library preparation, we used 1/4 \times and 1/5 \times reaction volumes of Nextera XT DNA Library Preparation Kit for scRR3 and scRR1, respectively. Meanwhile, the genomic DNA captured on the beads was processed in the same manner for both protocols (scRR3 and scRR1). The isolated DNA was subjected to whole-genome amplification (WGA) using SeqPlex Enhanced DNA Amplification

Kit (SEQXE), followed by library preparation using NEBNext® Ultra™ II DNA library Preparation kit for Illumina® or KAPA LTP Library Preparation Kit.

Reference:

Hayashi, T. et al. Single-cell full-length total RNA sequencing uncovers dynamics of recursive splicing and enhancer RNAs. *Nat. Commun.* 9, 619 (2018).

In addition, we also added a brief explanation regarding this point in the main text as follows:

[Page 5]

“We established two slightly different versions of the scRR-seq protocol, scRR1 and scRR3, using initial cell lysis buffer volumes of 1 μ l and 3 μ l, respectively. **These two versions were developed in parallel, as each offers distinct advantages depending on the experimental context. scRR1 is more cost-effective and well-suited for large-scale experiments where sample preparation is straightforward and a certain degree of failure is acceptable, such as when single cells are sorted by flow cytometry. In contrast, scRR3 is more appropriate for manually isolated samples that are difficult to collect, have larger carryover volumes, or are otherwise precious and require a high success rate (Supplementary Note 1 and Supplementary Fig. 2).”**

2. Performing FACS is challenging due to the limited number of cells at the 8-cell stage, and relying solely on HMM analysis makes it difficult to distinguish whether cells are in the G1 or G2/M phase. How was the cell cycle of eight-cell evaluated? Was this assessment based on karyotype or other methods?

Thank you for the question.

To evaluate the cell cycle state of individual cells in 8-cell stage embryos, we performed karyotype analysis based on scRR-seq-DNA data. We applied 6-state HMM calling using AneuFinder with 500-kb bins to analyze CNV states. Karyograms were generated and used to infer the cell-cycle stage of each cell. Cells showing alternating one- and two-copy number states across the genome were interpreted as being in S phase (as they reflect unreplicated and replicated DNA, respectively), while cells with a uniform copy number state across the genome were considered to be in either G1 or G2/M phase.

In our dataset, most cells from embryo #1 showed uniform copy number profiles, while most cells from embryo #2 exhibited one- and two-copy patterns consistent with the cells being in S phase (Fig. 2 for Reviewers). We added representative karyogram profiles to show these classifications to the new Supplementary Fig. 4.

As the reviewer pointed out, it is challenging to distinguish between G1 and G2/M phases based solely on copy number, as both can exhibit similar copy number profiles (e.g., Supplementary Fig. 4a). We acknowledge this point and have revised the manuscript to refer to these cells as “G1-like and G2M-like cells” rather than assigning them definitively to G1 cells, in order to improve the accuracy. We revised the text as follows:

[Page 8]

“We obtained **G1-like and G2/M-like cells** from 8-cell embryos (Supplementary Fig. 4a), which were used to correct the mappability of S-phase cells (Supplementary Fig. 4b). We successfully generated scRT profiles of 8-cell embryos from scRR-seq-DNA at 40-kb resolution, which was challenging for G&T-seq (Fig. 2d), possibly due to its low coverage (Extended Data Fig. 4f and Supplementary Table 1). Taken together, scRR-seq showed enhanced resolution and sensitivity compared to G&T-seq for both DNA-seq and RNA-seq, especially the former.”

Fig. 2 for Reviewers. Karyograms of cells isolated from two 8-cell embryos.

a) Most cells from embryo #1 exhibited uniform copy number profiles, with the exception of a few regions showing chromosomal abnormalities. b) Most cells from embryo #2 showed a mixture of one- and two-copy patterns, suggesting that these cells were undergoing DNA replication during the S phase.

3. When concluding that "human Xi's RT is less uniformly late compared to the mouse Xi" (line 278 in the main text and line 1155 in Supplementary Note 2), the authors do not reference any figure or statistical results. Furthermore, no data on the mouse Xi are presented in this study. The basis for this conclusion is unclear.

We appreciate the reviewer's comments and apologize for the lack of supporting data in the previous version of the manuscript. In the revised manuscript, we have included new results comparing the single-cell replication timing profiles of human and mouse Xi and Xa, shown in the newly added Supplementary Fig. 7. We also added explanation and discussion of these results in Supplementary Note 2, as follows:

[Pages 46–47]

“Haplotype-a of chrX, which most likely corresponds to the inactive X chromosome (Xi), exhibited distinct replication behavior. Unlike the mouse Xi in neural stem cells, which displays rapid and uniform late replication (Supplementary Fig. 7a; Poonperm et al., *Nat. Struct. Mol. Biol.* 2023), the human Xi in hTERT-RPE1 cells did not (Supplementary Fig. 7a). This observation led us to hypothesize that the human Xi exhibits less uniform late replication than the mouse Xi.

To test this, we developed a simple metric, the DNA copy number uniformity score, to quantify the consistency of replication status across bins within a chromosome in individual cells (Supplementary Fig. 7b,c). The logic is as follows: chromosomes that replicate uniformly at a given time point should exhibit similar replication status across most bins, resulting in a higher score. For example, in early S-phase (most regions are unreplicated) and late S-phase (most regions are fully replicated), uniformity is high. In contrast, during mid S-phase, when replication is already completed in some bins/regions while others remain unreplicated, uniformity is reduced.

Applying this analysis, we found that the mouse Xi followed the expected pattern: high and stable uniformity during early to mid S phase (up to ~60% replication score), followed by a sharp drop to lower uniformity in mid to late-S (~60–80% replication score), before quickly recovering to high uniformity in late-S phase. By contrast, the human Xi exhibited persistently lower copy number uniformity scores across all stages of S-phase, suggesting that replication on the human Xi is more variable (Supplementary Fig. 7d). We also performed the same analysis on the Xa in both species. In both human and mouse cells, the Xa exhibited somewhat similar DNA copy number uniformity patterns: higher uniformity during early and late S phases, and lower uniformity in mid S, although the pattern was less clear in human cells (Supplementary Fig. 7d).

While this may reflect true biological differences, technical factors such as SNP density or mappability could also contribute to these observations. Nonetheless, these results again underscore the limitations of population-based haplotype-specific BrdU-IP data, which could not clarify these differences. These results highlight the advantage of using scRepli-seq data.”

4. In lines 157-162 on pages 6-7, why are non-poly(A) RNAs detected as efficiently as with scRamDA-seq, but lncRNA detection efficiency is lower? What is the preference of this technique for detecting RNAs? Is the detection efficiency for all such RNAs the same except well-known non-poly(A) RNAs?

Thank you for your comments. We agree that these points require further clarification. To address them, we performed additional analyses and experiments. The new results are presented in Extended Data Fig. 3d,e and Supplementary Fig. 3. Specific responses to your questions are provided below:

In principle, RamDA-seq and scRR-seq-RNA can capture both poly(A)+ and non-poly(A) RNAs, as they utilize a combination of oligo(dT) primers and not-so-random primers for reverse transcription (Hayashi et al., *Nat. Commun.* 2018). Therefore, this approach enables broad RNA detection, covering both coding and non-coding RNAs.

Non-poly(A) RNAs include a diverse group of transcripts, such as certain histone mRNAs and lncRNAs. As shown in Fig. 1f, scRR-seq-RNA could detect well-known non-poly(A) RNAs with efficiency comparable to scRamDA-seq. We reasoned that this is because these transcripts tend to be relatively abundant and stable (e.g., 7 out of 9 genes shown in Fig. 1f have $\log_{10}(\text{median TPM}+1) > 2.19$; Table 2 for Reviewers). This abundance likely facilitates their reverse transcription and capture.

Table 2 for Reviewers: Gene expression level of non-poly(A) RNAs shown in Fig. 1f.

Gene name	median TPM	mean TPM	log10(median TPM+1)	log10(mean TPM+1)
RN7SK	2897.519689	3346.284777	3.462176255	3.448014503
RN7SL1	2219.53313	3274.988533	3.346457257	3.397888306
MALAT1	1273.532764	1666.010486	3.105351004	3.154153421
RNU4-1	1216.633423	1244.052242	3.085516561	3.038415483
NEAT1	502.5591023	561.3910335	2.702050451	2.597358058
RNU1-3	170.4486681	225.4919425	2.234134116	2.26353915
RMRP	154.3924839	170.9418589	2.191430009	2.180237742
RPPH1	53.36841365	62.4031994	1.735346661	1.652255202
TERC	1.542916624	2.845900149	0.405332121	0.407980159

Nonetheless, as the reviewer mentioned, many lncRNAs were not efficiently detected by scRR-seq-RNA. One reason is that they generally have low expression levels. As shown in Extended Data Fig. 3a, genes detected only in scRamDA-seq but not scRR-seq show lower expression levels than those that can be detected by both methods. However, we hypothesized that other factors, such as subcellular localization, might also contribute. This is because scRR-seq involves the separation of DNA and RNA before profiling. RNAs associated with chromatin or retained in the nucleus (i.e., on beads) may be underrepresented in the cytoplasmic (supernatant) fraction. To test this, we utilized published RNA localization data from several human cell lines (Mas-Ponte et al., *RNA*. 2017) to examine the potential subcellular localization of RNAs detected only by RamDA-seq and those detected by both methods. Indeed, we found that RNAs that were not detected by scRR-seq are more likely to show nuclear localization (Extended Data Fig. 3d,e).

To confirm this further, we performed additional experiments using RPE1 cells and the scRR1 protocol. After cell lysis, we collected RNA from the supernatant and the bead-bound fractions, which were subjected to scRR-seq-RNA and analyzed separately (Supplementary Fig. 3a).

Gene expression quantification by RSEM detected over 15,000 expressed genes (TPM>0) per cell in both supernatant-derived (cytoplasmic) and bead-derived (nuclear) samples (Supplementary Fig. 3b), with a slightly higher number observed in the bead-derived (nuclear) fraction. This indicated that certain RNAs were more enriched in the bead (nuclear) fraction. Clustering analysis identified a group of genes that shows a higher expression level in the bead fraction than the supernatant (Supplementary Fig. 3c, cluster 2). This group was significantly enriched for lncRNAs (Supplementary Fig. 3d). These results indicate that a subset of lncRNAs preferentially associates with the bead (nuclear) fraction.

Again, using published RNA localization data (Mas-Ponte et al., *RNA*. 2017), we observed that these bead-enriched RNAs (including lncRNAs) are often found in the nucleus (Supplementary Fig. 3e, cluster 2). We also found that this group of RNAs doesn't show very high expression levels (Supplementary Fig. 3c, cluster 2). These results were consistent with our earlier observation that a subset of genes was detected by scRamDA-Seq but not scRR-seq (Extended Data Fig. 3d,e).

In summary, our findings suggest that the inability of scRR-seq to detect some RNAs, including lncRNAs, is not due to its inability to detect certain types of RNAs. Rather, it is likely due to the characteristics of these RNAs, which tend to be lowly expressed and show nuclear localization. While RamDA-seq is designed to detect all types of RNAs without bias, scRR-seq-RNA detection efficiency varies depending on the RNA characteristics, such as abundance and subcellular localization.

To reflect these things, we revised the main text as follows:

[Pages 6–7]

“Our scRR-seq-RNA detected ~80% of genes and transcripts that were detected by the original scRamDA-seq (Fig. 1e and Extended Data Fig. 2g). scRR-seq-RNA also detected well-known non-poly(A) RNAs as efficiently as scRamDA-seq (Fig. 1f). The remaining ~20% not detected by scRR-seq-RNA were those that show low expression levels by scRamDA-seq and frequently corresponded to long non-coding (lnc)RNAs (Extended Data Fig. 3a–c). Using published RNA localization data²¹, we found that these undetected RNAs tend to be nuclear-localized (Extended Data Fig. 3d,e). This is consistent with the fact that, in scRR-seq-RNA, RNA was collected from the supernatant, which mainly corresponds to the cytoplasmic fraction. We further confirmed that a subset of RNAs, enriched for lncRNAs, remained bound to the beads (Supplementary Fig. 3). These results suggest that RNA abundance and subcellular localization influence detection sensitivity in scRR-seq-RNA. This could potentially explain the decrease in the number of genes detected compared to scRamDA-seq. Overall, however, scRR-seq-RNA data were comparable to scRamDA-seq data obtained individually.”

5. The discussion on the relationship between DNA copy number and gene expression is highly valuable and effectively showcases the capabilities of the authors' method. I have only one minor question here: as replication progresses, does the overall gene expression level in the cell increase? Could the absolute expression of 2-copy genes increase without being reflected in the $[\log_{10}(TPM+1)]$ values?

Thank you for your questions. Below, we responded to these questions one by one as follows:

(5.1) As replication progresses, does the overall gene expression level in the cell increase?

A recent study suggested that RNA content per cell increases by about twofold from the G1 to the G2/M phases (Riba et al., *Nat. Commun.* 2022; <https://doi.org/10.1038/s41467-022-30545-8>; see Fig. 3C in that paper). However, these dynamic changes appear to vary across different cell types. For instance, in mESCs, the increase occurs continuously, whereas in human fibroblasts, RNA levels tend to remain relatively stable from late G1 to early G2/M. Nonetheless, it is unclear whether the overall gene expression levels increase during the S phase.

To address this question, we used the human haploid HAP1 scRR-seq data collected from the whole-S phase. We measured the overall expression level of a gene by counting the number of sequencing reads mapped to the gene in a cell, based on the assumption that read counts reflect the transcription level (Conesa et al., *Genome Biol.* 2016). For a fair comparison, we scaled all scRR-seq-RNA data to 1 million reads per cell, then used featureCounts to quantify the number of reads mapped to each gene. Reads mapped to mitochondrial and ribosomal genes were removed from downstream analysis. To normalize the data, we used total reads from ERCC spike-in controls. These synthetic RNA molecules are added in equal amounts to all cells and serve as stable internal standards unaffected by the cell cycle stage. We then plotted the ERCC-normalized total mapped reads per cell against the percentage replication score (Fig. 3 for Reviewers) and applied linear fitting to capture the overall trend. Our results suggest that the total gene expression levels do not continuously increase during the S phase in HAP1 cells. Rather, they remain relatively stable, with higher cell-to-cell variability during mid-S phase. However, these trends may possibly differ in other cell types.

Fig. 3 for Reviewers. Overall gene expression per cell during S-phase progression.

ERCC-normalized total mapped reads per cell against the percentage replication score in HAP1 cells. Each point represents a single cell. The red line shows the linear fit.

(5.2) Could the absolute expression of 2-copy genes increase without being reflected in the $[\log_{10}(\text{TPM}+1)]$ values?

First, we apologize for the mistake in the previous figure label. We did not use TPM for this analysis. Actually, we directly used raw read counts from featureCounts, so the correct label in this figure and related information should have been “ $\log_{10}(\text{mapped reads} + 1)$ ” instead of “ $\log_{10}(\text{TPM} + 1)$.”

To validate this result, we repeated the analysis using the same HAP1 dataset, but this time, we adjusted all samples to the same library size (1 million reads per cell) to ensure a fair comparison of gene expression levels across samples. We examined both raw mapped counts and \log_{10} -normalized values (Fig. 4 for Reviewers). To provide a more detailed view, we categorized gene expression patterns into five groups as shown below (Fig. 4 for Reviewers). We found that ~58% of spliced RNAs had higher median expression in the 2-copy gene group, while ~36% showed higher expression in the 1-copy gene group. However, most of these differences were not statistically significant. Only ~9% of spliced RNAs showed significant differences in gene expression levels between the two groups. For unspliced RNAs, ~48% had higher median expression in the 2-copy gene group, and ~40% showed higher expression in the 1-copy gene group. Similarly, ~7% of unspliced RNAs showed statistically significant differences.

We have updated the manuscript with these new results using $\log_{10}(\text{mapped reads} + 1)$ as shown in Fig. 5b,d, and Supplementary Table 4.

Fig. 4 for Reviewers. Gene expression patterns of replicated (2-copy) and unreplicated (1-copy) genes during mid-S phase in HAP1 cells.

Pie charts show five categories of gene expression patterns. Both mapped reads and $\log_{10}(\text{mapped reads} + 1)$ were used to assess median gene expression levels and perform statistical analysis. The Wilcoxon test was used for statistical comparison. Numbers on the charts indicate the percentage of genes in each category, and the number below each chart shows the total number of informative genes analyzed.

6. It would also be helpful if the authors could discuss the cost and time efficiency of their method compared to other techniques in the Discussion section.

Thank you for this helpful suggestion. We have summarized the cost of scRR-seq in a new Supplementary Table 5. Also, we added a paragraph in the Discussion section to address the cost and time efficiency of our method in comparison with other single-cell multi-omics techniques, as follows:

[Page 17]

“Regarding the overall cost and time efficiency, we believe scRR-seq is comparable to most existing protocols. It typically requires approximately 2–4 days each for DNA-seq and RNA-seq library preparation, which is similar to standard NGS workflows. The scRR-seq protocol is simple and easy, as it utilizes the commercially available RamDA-seq kit for RNA-seq. This simplifies the workflow and reduces variability associated with manual protocols. In terms of cost, our optimized scRR-seq protocol lowers reagent expenses to about \$60 per sample for combined DNA and RNA-seq library preparations (see Supplementary Table 5). In comparison, most other methods use Smart-seq-based technology for the RNA-seq component, which either requires manual protocols or commercial kits that are substantially more expensive (which cost ~\$13 and ~\$80/sample for RNA-seq alone, respectively⁴⁰). Although scRR-seq may not provide a major advantage in processing time or total cost, its simplified workflow makes it more user-friendly and easier to adopt in standard laboratory settings.”

7. Line 260: “We obtained a trajectory of S-phase progression relatively similar to CBMS1 mESCs (Extended Data Fig. 5a)” should instead reference Extended Data Fig. 5d.

Thank you for pointing this out. We have corrected the figure reference as suggested. As we have modified the figures, the sentence now correctly refers to Fig. 3d.

8. In Fig. 4d, is the scRT shown for chr4 or chrX? The title and caption indicate chr4, but the genomic range in the figure is labeled as “chrX:1-60.”

Thank you for pointing this out. We apologize for the mistake in the title and caption, which incorrectly referred to chrX. The scRT data in Fig. 4d is indeed from chr4, and the genomic label should be chr4:1–60.

We have corrected the label in the revised version to accurately reflect that the data shown corresponds to chr4.

Reviewer #3 (Remarks to the Author):

In the manuscript, Poonperm et, al. developed the single-cell Repli-RamDA-seq, also called scRR-seq. As a multi-omics co-assay tool, sRR-seq can be applied to profile high-resolution DNA replication and quantify full-length total RNA within the same cell. Specifically, scRR-seq has shown promising performance regarding the DNA-seq and RNA-seq quality when compared to scRepli-seq, scRamDA-seq, and G&T-seq. The authors have also applied scRR-seq to study the dynamics of the cell cycle on transcriptome and found that S-phase-associated genes are cell type-specific, and a uniform cell cycle marker set can not define cell cycle status accurately. Moreover, they have explored the effect of DNA replication on gene expression. Overall, scRR-seq can be a powerful tool for studying cells from the perspective of genome and transcriptome. I also have a few comments:

We appreciate the reviewer for the positive feedback on our study. We have addressed the reviewer's specific questions and comments in detail below.

Major comments:

1. It is encouraged to release all code for the data analysis of this paper to ensure the result is reproducible. Also, the sequencing data can not be found through GEO.

Thank you for the suggestions. To support reproducibility, the code used for data analysis will be made publicly available on GitHub.

GitHub link: <https://github.com/mcbmieu/scRR-seq>

As the repository is not yet publicly accessible, we have deposited the analysis code on Figshare for your review. It can be accessed using the following link.

Figshare link: <https://figshare.com/s/f7b3fba5caa1657176fd>

We apologize for the issue with accessing our sequencing data. The dataset has been deposited in GEO and can be accessed using the following link and token:

GEO link: <https://www.ncbi.nlm.nih.gov/geo/query/acc.cgi?acc=GSE278959>

Access token: khelqgendqjtm

These details have also been included in the Data Availability section of the revised manuscript.

2. I noticed that you have cited DR-seq without benchmarking with it. Is there any specific reason for that?

Thank you for your question. We cited DR-seq to acknowledge its contribution as a related method, but we did not include a direct comparison with our scRR-seq method for the following reasons:

- **Limited data availability:** DR-seq was performed on two different cell lines: E14 mESCs and the human breast cancer cell line SK-BR-3. Of these, only the E14 mESC data are applicable to our study, as we also used mESCs. However, DR-seq provides DNA data for only three E14 mESC samples, which we considered insufficient for a fair or reliable comparison.

- **Technical differences in the DNA-seq component:** The way DR-seq processes RNA and DNA differs substantially from our method. In DR-seq, RNA and DNA are not physically separated but are instead processed together in a single reaction that is subsequently split for sequencing. This design introduces several analytical challenges distinct from those encountered in our approach. For example:
 - (i) Exclusion of DNA reads: DR-seq has difficulty distinguishing DNA reads that overlap coding regions, as such reads could originate from either cDNA or genomic DNA. To address this, the original authors excluded reads overlapping coding regions from their DNA-seq analysis. Although they noted that these regions account for only ~2% of the genome and do not impact large CNV detection, this exclusion could alter the genome representation and complicate downstream comparison for high-resolution copy number analysis that we use to assess DNA replication status.

 - (ii) Unequal genomic binning: Due to the masking of coding regions, DNA reads in DR-seq are assigned to genomic bins of unequal size. This is incompatible with our scRR-seq-DNA pipeline, which relies on consistent, high-resolution binning to accurately analyze replication state and CNVs.

 - (iii) Amplification biases: DR-seq uses quasilinear amplification, which introduces specific biases. Although these biases were corrected using a custom coverage-based model developed by the DR-seq authors, such corrections are not directly transferable to our data or analytical framework.

As an example, we plotted read coverage of DNA-seq from DR-seq (DR-seq-DNA) in E14 mESCs along chromosome 5 and compared with reads per million (rpm) of our scRR-seq-DNA (Fig. 5a,b

for Reviewers). Since no cell-cycle information is available for the DR-seq cells, we assumed that they were in S-phase, as we observed higher and lower coverage reads in early-replicating and late-replicating regions, respectively, (Fig. 5b,c for Reviewers; upper panel shows population-based mESC RT profile). However, the read distribution within those regions is uneven (Fig. 5b for Reviewers). This unevenness may result from: (1) a mixture of replicated (high copy) and unreplicated (low copy) DNA within the same region; (2) exclusion of reads overlapping with cDNA; or (3) normalization procedures correcting for amplification bias. In contrast, our scRR-seq-DNA from S-phase cells shows a much more uniform read distribution, with higher reads in early-replicating regions and lower reads in late-replicating regions (Fig. 5b,d for Reviewers). As a result, when we performed log₂median transformation on our scRR-seq-DNA data, the profiles closely resembled the RT profile of population mESCs (Fig. 5a,b for Reviewers; upper panel shows population-based mESC RT profile), while DR-seq data show less clear results. To quantify the variability of data, we calculated the median absolute deviation (MAD) across the genome (Fig. 5e for Reviewers), and across early- and late-replicating regions (Fig. 5f for Reviewers). DR-seq-DNA showed higher MAD scores in all regions analyzed, indicating high variability and less uniform coverage.

In summary, while DR-seq may be applicable for general CNV detection, we believe it is less appropriate for analyzing high-resolution replication timing based on copy number analysis. Taken together, we considered that comparing the DR-seq and scRR-seq directly may give a misleading impression, rather than a fair evaluation.

Fig. 5 for Reviewers. DNA-seq read distribution of DR-seq and scRR-seq in mouse ESCs.

a) Read coverage along chromosome 5 (chr5) from DR-seq-DNA in three E14 mESCs and read count per million (rpm) from scRR-seq-DNA in three CBMS1 mESCs. The top panel shows BrdU-IP Repli-seq (population-based) RT profiles from JB4 mESCs (Poonperm et al., *Nat. Struct. Mol. Biol.* 2023) for a comparison. Yellow and blue represent early and late RT domains, respectively. Log2median profiles from scRR-seq-DNA closely resemble the population-based RT profiles, whereas those from DR-seq-DNA appear less defined. b) Zoomed-in view of the

region shown in a). e) MAD scores calculated for DR-seq-DNA and scRR-seq-DNA data across the whole genome for each single cell. f) MAD score calculated for DR-seq-DNA and scRR-seq-DNA across early- and late-replicating regions for each single cell. Bins with BrdU-IP Repli-seq RT value > 0.25 were classified as Early-RT, and $RT < -0.25$ as Late-RT.

3. As far as I know, it will largely increase the scope of scRR-seq when you apply it to cancer studies, such as the relationship between CNV, mutations, and gene expression. Will it be able to provide one additional figure for such a case study?

We greatly appreciate the reviewer's insightful suggestion to explore the application of scRR-seq in cancer research. We fully agree that combining scRR-seq with CNV, mutation, and gene expression analysis in cancer cells would significantly enhance the impact and utility of this technology. However, due to limitations in sample availability and resources, we are currently not in a position to include a cancer-focused case study in the present work. While we understand that such an application would broaden our work, we feel that it would be a bit beyond the intended scope of this manuscript.

In response to this suggestion, we have added a perspective on how scRR-seq could be applied to cancer biology in the Discussion section as follows:

[Page 17]

“Since the regulation of gene expression associated with CNVs is complex and not fully understood³⁸, scRR-seq offers a unique opportunity to explore this relationship at the single-cell level. **In particular, scRR-seq could be highly useful in cancer research, where gene expression and CNVs vary between cells and influence tumor progression and therapeutic response³⁹.** Furthermore, we recently demonstrated that scRepli-seq can identify CNVs in mouse embryos at the single-cell level, regardless of whether cells are in G1 or S-phase¹⁶. This suggests that scRR-seq could be extended to explore gene expression and CNV associations in S-phase cells, which would be particularly valuable for analyzing rare or limited samples, for instance, single cells derived from tissues *in vivo* **or patient samples.**”

4. The comparison of identifying S phase cells between the Seurat gene set and SRR-seq is unfair. Will it also be able to generate the data from other known phases as ground truth and build a model to predict cell-cycle status using scRR-seq? The model can be very simple, such as a linear model. Then it is possible to compare the accuracy of the two methods for identifying cell cycle status.

Thank you for your valuable feedback. First, we apologize for any confusion caused by our original explanation. Our primary focus in this study is not on comprehensive cell cycle classification, but rather on capturing gene expression dynamics during S-phase progression using scRR-seq. This is because one of the key advantages of scRR-seq is its ability to simultaneously assess transcriptional and replication dynamics within S-phase, which are difficult using conventional gene expression-based methods.

We agree with the reviewer that the original comparison between the Seurat cell-cycle gene set and scRR-seq S-phase progression markers was not fully fair. The Seurat gene set is meant to broadly classify cells into G1, S, and G2/M phases using genes that are specifically expressed in S- and G2/M-phases. In contrast, our S-phase progression markers were developed based on data derived from only S-phase cells and are intended to resolve cell states only within the S phase, not across the entire cell cycle. These differences in purpose and scope have introduced bias in the comparison. In addition, our original comparison may have also created the misleading impression that our newly identified markers were intended for full cell cycle classification, and vice versa. To address these issues, we have removed the comparison between the Seurat cell-cycle gene set and scRR-seq's S-phase progression markers (original Fig. 3e,f, middle panels) from the revised manuscript.

We also appreciate the suggestion to build a predictive model for cell-cycle phases using scRR-seq data. However, one limitation of scRR-seq-DNA (and scRepli-seq), as currently implemented, is that we cannot clearly distinguish between G1 and G2/M phases, due to the challenge of resolving these states based solely on DNA copy number. While our HMM-based DNA replication analysis is well-suited for detecting progression through S-phase, we cannot confidently assign G1 or G2/M phases. For instance, as noted in our responses to Reviewer 2's comment #2, we cannot distinguish whether the cells from embryo #1 are in G1 or G2/M phase. As such, building a full cell-cycle phase prediction model based solely on scRR-seq would be limited.

In response to your comments, we have revised the manuscript to tone down our claims about full cell-cycle identification and instead emphasize the strength of our method in capturing detailed S-phase progression and identifying novel S-phase progression markers, which, to our knowledge,

have not been reported so far. We have rewritten the original section titled “*scRR-seq reveals cell-cycle gene expression dynamics at high resolution*” and divided it into two sub-sections. We first highlight the ability of scRR-seq to identify new markers associated with S-phase progression. Then, we discuss the limitations of existing cell-cycle markers and assignment methods. We hope this revised version more accurately reflects the scope and strengths of our method.

The revised text is as follows:

[Pages 9–11]

scRR-seq captures gene expression dynamics and novel markers during S-phase progression

As scRT data allows one to tell the S-phase timepoint of each cell (Supplementary Fig. 5), we reasoned that scRR-seq data should reveal gene expression dynamics during **S-phase progression** at an unprecedented temporal resolution **and may help identify S-phase progression markers, which have never been reported before**. Therefore, we generated scRR-seq data from hTERT-RPE1 cells and CBMS1 mESCs throughout the S-phase. First, we used scRR-seq-DNA to generate a ‘whole-S’ RT profile, which showed that we successfully captured the entire S-phase of hTERT-RPE1 cells (Fig. 3a) and CBMS1 mESCs (Fig. 3b and Extended Data Fig. 5). The whole-S RT profile of CBMS1 mESCs closely resembled the profile obtained previously by scRepli-seq¹³ (Extended Data Fig. 6a). Then, we used our hTERT-RPE1 and CBMS1 mESC scRR-seq-RNA data to find novel S-phase progression marker candidates. We sorted cells based on their percentage replication scores and searched for genes that showed significant gene expression changes during S-phase progression. Using the generalized additive model (GAM) fitting, we identified 52 significant dynamic genes (FDR<0.05) in hTERT-RPE1 and 55 genes in CBMS1 mESCs (FDR<0.01) and divided them into three groups based on their expression patterns (Extended Data Fig. 6b,c and Supplementary Table 2). Gene Ontology (GO) analysis revealed that these newly identified genes were significantly overrepresented in processes related to cell division (Supplementary Table 2). Only a few genes were shared between the two cell lines (Supplementary Table 2).

Utilizing these newly identified genes for diffusion map analysis (DMA), we observed a clear trajectory of S-phase progression (Fig. 3c). However, these new human **S-phase progression** markers were unable to reveal the S-phase progression trajectory in mice and vice versa, suggesting the species- or cell-type-specificity of markers (Fig. 3c, **Supplementary Fig. 6a–e**). Lastly, we applied these newly identified S-phase progression markers of CBMS1 to another mESC scRNA-seq dataset¹⁴ to assess their reproducibility. We obtained a trajectory of S-phase progression relatively similar to that of CBMS1 mESCs (Fig. 3d), confirming the robustness of these markers. Taken

together, our results demonstrate that scRR-seq enables the comprehensive analysis of gene expression dynamics during **S-phase progression** at an unprecedented temporal resolution. Moreover, it also allows the identification of a novel set of S-phase progression markers for a given cell type and species.

scRR-seq reveals a limitation of existing cell-cycle stage assignment

Cell-cycle markers are commonly used to assign cell-cycle stages to scRNA-seq datasets. For example, Seurat²³, a widely used scRNA-seq analysis pipeline, employs a predefined set of S and G2/M markers from human cells²⁴ to classify cells into specific cell-cycle stages. Since our scRR-seq datasets capture both the S-phase state (Supplementary Fig. 5) and gene expression profiles, we leveraged this dual capability to assess the accuracy of existing cell-cycle markers and cell-cycle stage assignments.

We used Seurat and its default cell-cycle marker sets to assign cell-cycle stages to individual cells in our whole-S-derived hTERT-RPE1 and CBMS1 mESCs scRR-seq datasets, based on the scRR-seq-RNA data. We then compared these assignments with the percentage replication scores derived from the corresponding scRR-seq-DNA data.

Unexpectedly, while all cells were in S-phase based on the percentage replication scores, Seurat assigned G1, S, and G2/M phases to these cells (Fig. 3e,f). In hTERT-RPE1 cells, while the S and G2/M phase assignments were relatively reasonable and marked cells with low and high percentage replication scores, respectively, some cells were incorrectly labeled as G1 (Fig. 3e).

In contrast, in CBMS1 mESCs, the majority of cells were assigned as G1 (Fig. 3f). The proportion of cells assigned as S and G2/M was noticeably low, which corresponded to those with low (<40%) and high (>70%) replication scores, respectively (Fig. 3f). Given that Seurat's default markers are derived from human tumor cells²⁴, our findings suggest that cell-cycle markers may be cell-type or species-specific (Supplementary Fig. 6f). Overall, these results underscore a limitation of current markers or methodology for cell-cycle stage assignment, particularly when applied across different cell types or species.

Minor comments:

5. The comparison between scRR-seq to G&T-seq is not fully fair. For example, in the DNA part, scRR-seq-DNA adjusted reads to about 4M/sample. It should also consider cell number and make sure the similar level of mean read count per cell. This is to address my concern about the potential large difference in cell number per sample between SRR-seq and G&T-seq.

Thank you for your suggestions. To address your concern, we improved the comparison between scRR-seq and G&T-seq by separating the G&T-seq samples into two groups based on their initial read counts: one with higher read counts than our samples [referred to as G&T-seq(H)], and another with lower read counts [referred to as G&T-seq(L)] for both DNA-seq and RNA-seq components. We then repeated the comparison, as shown in the **updated Fig. 2b** and **Extended Data Fig. 4**. We found that, regardless of the initial read counts, the DNA-seq component of G&T-seq still did not achieve the same high coverage as our scRR-seq-DNA. Similarly, the RNA-seq results from both G&T-seq(H) and G&T-seq(L) were consistent with each other, showing lower transcript and gene detection compared to scRR3, but comparable to scRR1 in this experiment. These results support our original conclusion that scRR-seq shows better performance over G&T-seq under the tested conditions.

6. Please also provide additional support for your saying that cell-cycle markers are cell-type-specific. In your provided evidence, the difference may be introduced by different species.

Thank you for this suggestion. We agree that species differences could have contributed to the observed variation, and we appreciate the opportunity to provide additional support for our claim.

We would like to clarify that our current focus is specifically on S-phase progression markers, rather than general cell-cycle markers across all phases. To address your concern, we have now included new data on S-phase progression markers in human-derived HAP1 cells, which are shown in the new Supplementary Fig. 6a–e. These new results support the idea that S-phase progression markers can vary between different cell types, even within the same species, and also suggest that S-phase progression markers are cell-type-specific.

In addition, we also used Seurat and its default cell-cycle markers and applied them to our HAP1 scRR-seq-RNA data (new Supplementary Fig. 6f). Similar to what was observed in hTERT-RPE1 (Fig. 3e), despite HAP1 cells being in S-phase based on the percentage replication scores, Seurat

assigned them to G1, S, and G2/M phases. While the S and G2/M phase assignments were relatively reasonable and marked cells with low and high percentage replication scores, respectively, some cells were assigned as G1. Taken together, these results suggest that cell-cycle markers may be cell-type or species-specific.

7. It is not surprising that DNA copy numbers can not influence gene expression in normal cells. As I have mentioned in major comments, the relationship between CNV and gene expression in cancer cells can be more interesting.

Thank you for your comment. As mentioned in our response to comment 3, we agree that the relationship between CNVs and gene expression in cancer cells is particularly interesting, and we also recognize the potential of scRR-seq in this area. Therefore, we have expanded the Discussion section to highlight its possible application to cancer research.

Reviewer #4 (Remarks to the Author):

We appreciate the reviewer for their time and thoughtful contribution to the review process.

Point-by-point response to all of the reviewers' criticisms (reviewer comments in bold italic):

REVIEWER COMMENTS

Reviewer #2 (Remarks to the Author):

In this work the authors developed a single-cell multi-omics method by physically separating DNA and RNA using magnetic beads, enabling the integration of scRepli-seq and scRamDA-seq. This new approach achieves comparable performance to the original protocols for both modalities. This method demonstrates several notable strengths. It effectively identifies the cell cycle stages of S-phase cells and discovers new S-phase progression markers, which is particularly significant since the cell cycle is a common confounding factor in scRNA-seq data, and existing annotation methods remain suboptimal. Moreover, the authors provide genome-wide analysis of how changes in gene copy number during replication influence gene expression levels, a capability uniquely enabled by single-cell techniques.

We would like to thank the reviewer for the positive feedback on our study. We have addressed the reviewer's specific questions and comments in detail below.

Below are some specific comments:

1. The article provides two protocols: scRR1 and scRR3, mainly differing in lysis buffer volume. Why were these two protocols designed separately, and what application scenarios is each suitable for? In the scRR1 reaction, only 1 µl of the 5 µl reverse transcription mixture was used for further reactions. Why is this? Does this step result in RNA loss and lead to experimental dropout?

Thank you for your questions. Below, we address questions separately for clarity:

(1.1) Why were these two protocols designed separately, and what application scenarios is each suitable for?

The scRR3 and scRR1 were developed based on the original RamDA-seq protocol, which includes two recommended lysis volumes: a standard protocol using 3 µl and an optional protocol using 1 µl. Initially, the 3 µl protocol was established and provided robust RNA detection. Then, to enhance the workflow and reduce costs, especially for processing large numbers of samples at once, the 1 µl protocol was established and became the basis for scRR1.

The 1 μ l (scRR1) protocol improves the workflow as follows:

- **Reduced reagent costs:** As reaction volumes are scaled down, the cost is \sim 3 times reduced compared to scRR3 (see Supplementary Table 5).
- **Simplified workflow:** The second-strand synthesis products (cDNA) can be used directly for Nextera XT library preparation without the DNA purification step (Supplementary Fig. 2). This saves hands-on time and reduces the risk of sample loss.
- **Minimized risk of library failure:** In the 3 μ l protocol, DNA purification using AMPure XP is required before library preparation (Supplementary Fig. 2). This includes a washing step with ethanol, followed by elution of DNA (see details in Method, section “Library preparation for scRR-seq-RNA”). However, ethanol contamination is critically problematic. If any ethanol remains in this step, even in small amounts, it can interfere with the library preparation reaction by Nextera XT in the next step and lead to failed or low-quality libraries. Since the 1 μ l protocol skips this purification step, it reduces this risk.

We observed that RNA detection is slightly better in samples prepared using the 3 μ l protocol compared to those prepared with the 1 μ l protocol (Fig. 1e). However, given its advantages in simplicity, cost-efficiency, and throughput, we believe the 1 μ l protocol is a more practical choice overall.

However, it is important to note that RamDA-seq lysis buffer is sensitive to excess carryover volume from cell isolation/sorting, which can negatively affect the downstream cDNA synthesis. While this is generally not an issue when single cells are sorted by flow cytometry, it could cause problems when picking single cells manually, such as when isolating cells from early embryos. For example, in the original 3 μ l protocol, the volume of the cell sample added to the lysis buffer should be less than 0.5 μ l (according to the RamDA-seq manual by TOYOBO). Proportionally, this would require the carryover volume to be less than 0.17 μ l in the 1 μ l protocol, which is technically difficult to achieve when single cells are picked manually. In such cases, it is better to use the 3 μ l protocol instead of the 1 μ l protocol to ensure robust results. Therefore, we provided two protocols, and the choice of protocol should be based on the specific experimental setup and objectives.

In summary, scRR3 is suited for experiments that involve difficult manual cell isolation, and/or when high-quality RNA-seq data is crucial, while scRR1 is best for single-cell sorting by flow cytometry and large-scale experiment setup. For easy comparison of the two protocols at a glance,

we provided a summary table for comparison in Supplementary Note 1. For your convenience, the table is also shown below (Table 1 for Reviewers).

Table 1 for Reviewers: Comparison of scRR1 and scRR3

Feature	scRR1	scRR3
Initial lysis buffer volume	1 μ l	3 μ l
Workflow complexity	Simple (no cDNA purification step required)	More steps (includes cDNA purification)
Hands-on time per library preparation	2–4 days	2–4 days
Cost per sample (DNA and RNA-seq)	Low (~\$60/sample)	High (~\$100/sample)
Compatibility with FACS	High	High
Compatibility with manual cell isolation	Low (requires the carryover volume to be <0.17 μ l)	High (requires the carryover volume to be <0.5 μ l)
Best suited for	Medium- to large-scale studies using FACS	Limited samples and manually isolated samples
Example use case	Cell lines, automation-friendly workflows	Embryo dissection, rare or precious samples
DNA detection sensitivity	High	High
RNA detection sensitivity	Slightly low (~3.5% lower than scRR3 for FACS samples; ~10% lower for manually isolated samples)	High

(1.2) Why is only 1 μ l of the 5 μ l reverse transcription mixture used in scRR1? Does this cause RNA loss or dropout?

In our scRR1 protocol, the Nextera XT library preparation reaction is scaled down to 1/5 \times volume to improve cost-effectiveness for single-cell applications. We found that 1 μ l is the maximum volume of cDNA (reverse transcription mixture) that can be added to this scaled-down reaction without interfering with its reaction efficiency. Importantly, this scaled-down setup does not reduce the success rate of library preparation.

Using 1 μ l out of 5 μ l of cDNA in scRR1 does not cause notable RNA loss or dropout. As shown in Fig. 1e, scRR1 detects only slightly fewer transcripts and genes than scRR3 (less than 5%

difference). This is because the RamDA-seq method amplifies cDNA during reverse transcription using the RT-RamDA approach, which can increase the cDNA amount by more than 10-fold in just 30 minutes (Hayashi et al., *Nat. Commun.* 2018, Fig. 1 for Reviewers). This strong amplification ensures that there is sufficient cDNA material for library preparation, even from a small input. As a result, scRR1 can capture global gene expression at the single-cell level, comparable to the scRR3 protocol.

Fig. 1 for Reviewers. Figure 1b from Hayashi et al., *Nat. Commun.* 2018 shows the relative yield of cDNA from mESCs using the RT-RamDA approach, which can increase cDNA yield by more than 10-fold in 30 minutes of reaction.

To improve the manuscript, we updated the experimental scheme and added a description of the library preparation procedures and the rationale behind them in Supplementary Figure 2 as follows:

Supplementary Figure 2: Schematic overview of scRR1 and scRR3 workflows

Supplementary Figure 2: Schematic overview of scRR1 and scRR3 workflows

Detailed protocols are described in the Methods section in the main text. Single cells were collected into 0.2-ml PCR tubes, followed by lysis using complete RamDA-seq lysis buffer (TOYOBO, RMD-101) containing Dynabeads® MyOne™ Carboxylic Acid (beads). If the initial RamDA-seq lysis buffer volume was 3 μ l, we refer to the protocol as scRR3, and if it was 1 μ l, we refer to it as scRR1. Since the RamDA-seq lysis buffer is sensitive to excess carryover during cell isolation/sorting, which can negatively impact downstream cDNA synthesis, it is recommended that the carryover volume of the cell sample added to the lysis buffer be less than 0.5 μ l for scRR3 (as specified in the RamDA-seq manual by TOYOBO). Proportionally, this corresponds to less than 0.17 μ l for scRR1.

The RNA and DNA fractions were separated using a magnetic stand, with genomic DNA being captured by the beads and RNA remaining in the solution. The RNA fraction was then processed for scRamDA-seq using GenNext®RamDA-seq™ Single Cell Kit (TOYOBO). Both protocols are relatively similar. After removing genomic DNA contamination, cDNA was directly amplified from RNA using not-so-random primers and strand displacement amplification (Hayashi et al., *Nat Commun.* 2018).

In the scRR3 protocol, a cDNA purification step using AMPure XP beads is required before library preparation. It is important to completely remove ethanol during this step, as residual ethanol can result in low library yields. However, in the scRR1 protocol, this purification step can be skipped, allowing direct library preparation. For scRR1, only 1 μ l (1/5 \times of cDNA product) is used for library preparation, as this is optimized for the 1/5 \times scaled-down Nextera XT DNA Library Preparation Kit. Despite using a small volume, there is no notable drop in data quality or gene detection. As shown in Fig. 1e, scRR1 detects only slightly fewer transcripts and genes than scRR3 (less than 5% difference). This is because the RamDA-seq method amplifies cDNA during reverse transcription using the RT-RamDA approach, which can increase the cDNA amount by more than 10-fold in just 30 minutes (Hayashi et al., *Nat. Commun.* 2018). This strong amplification ensures that there is sufficient cDNA material for library preparation.

For library preparation, we used 1/4 \times and 1/5 \times reaction volumes of Nextera XT DNA Library Preparation Kit for scRR3 and scRR1, respectively. Meanwhile, the genomic DNA captured on the beads was processed in the same manner for both protocols (scRR3 and scRR1). The isolated DNA was subjected to whole-genome amplification (WGA) using SeqPlex Enhanced DNA Amplification

Kit (SEQXE), followed by library preparation using NEBNext® Ultra™ II DNA library Preparation kit for Illumina® or KAPA LTP Library Preparation Kit.

Reference:

Hayashi, T. et al. Single-cell full-length total RNA sequencing uncovers dynamics of recursive splicing and enhancer RNAs. *Nat. Commun.* 9, 619 (2018).

In addition, we also added a brief explanation regarding this point in the main text as follows:

[Page 5]

“We established two slightly different versions of the scRR-seq protocol, scRR1 and scRR3, using initial cell lysis buffer volumes of 1 μ l and 3 μ l, respectively. **These two versions were developed in parallel, as each offers distinct advantages depending on the experimental context. scRR1 is more cost-effective and well-suited for large-scale experiments where sample preparation is straightforward and a certain degree of failure is acceptable, such as when single cells are sorted by flow cytometry. In contrast, scRR3 is more appropriate for manually isolated samples that are difficult to collect, have larger carryover volumes, or are otherwise precious and require a high success rate (Supplementary Note 1 and Supplementary Fig. 2).”**

2. Performing FACS is challenging due to the limited number of cells at the 8-cell stage, and relying solely on HMM analysis makes it difficult to distinguish whether cells are in the G1 or G2/M phase. How was the cell cycle of eight-cell evaluated? Was this assessment based on karyotype or other methods?

Thank you for the question.

To evaluate the cell cycle state of individual cells in 8-cell stage embryos, we performed karyotype analysis based on scRR-seq-DNA data. We applied 6-state HMM calling using AneuFinder with 500-kb bins to analyze CNV states. Karyograms were generated and used to infer the cell-cycle stage of each cell. Cells showing alternating one- and two-copy number states across the genome were interpreted as being in S phase (as they reflect unreplicated and replicated DNA, respectively), while cells with a uniform copy number state across the genome were considered to be in either G1 or G2/M phase.

In our dataset, most cells from embryo #1 showed uniform copy number profiles, while most cells from embryo #2 exhibited one- and two-copy patterns consistent with the cells being in S phase (Fig. 2 for Reviewers). We added representative karyogram profiles to show these classifications to the new Supplementary Fig. 4.

As the reviewer pointed out, it is challenging to distinguish between G1 and G2/M phases based solely on copy number, as both can exhibit similar copy number profiles (e.g., Supplementary Fig. 4a). We acknowledge this point and have revised the manuscript to refer to these cells as “G1-like and G2M-like cells” rather than assigning them definitively to G1 cells, in order to improve the accuracy. We revised the text as follows:

[Page 8]

“We obtained **G1-like and G2/M-like cells** from 8-cell embryos (Supplementary Fig. 4a), which were used to correct the mappability of S-phase cells (Supplementary Fig. 4b). We successfully generated scRT profiles of 8-cell embryos from scRR-seq-DNA at 40-kb resolution, which was challenging for G&T-seq (Fig. 2d), possibly due to its low coverage (Extended Data Fig. 4f and Supplementary Table 1). Taken together, scRR-seq showed enhanced resolution and sensitivity compared to G&T-seq for both DNA-seq and RNA-seq, especially the former.”

Fig. 2 for Reviewers. Karyograms of cells isolated from two 8-cell embryos.

a) Most cells from embryo #1 exhibited uniform copy number profiles, with the exception of a few regions showing chromosomal abnormalities. b) Most cells from embryo #2 showed a mixture of one- and two-copy patterns, suggesting that these cells were undergoing DNA replication during the S phase.

3. When concluding that "human Xi's RT is less uniformly late compared to the mouse Xi" (line 278 in the main text and line 1155 in Supplementary Note 2), the authors do not reference any figure or statistical results. Furthermore, no data on the mouse Xi are presented in this study. The basis for this conclusion is unclear.

We appreciate the reviewer's comments and apologize for the lack of supporting data in the previous version of the manuscript. In the revised manuscript, we have included new results comparing the single-cell replication timing profiles of human and mouse Xi and Xa, shown in the newly added Supplementary Fig. 7. We also added explanation and discussion of these results in Supplementary Note 2, as follows:

[Pages 46–47]

“Haplotype-a of chrX, which most likely corresponds to the inactive X chromosome (Xi), exhibited distinct replication behavior. Unlike the mouse Xi in neural stem cells, which displays rapid and uniform late replication (Supplementary Fig. 7a; Poonperm et al., *Nat. Struct. Mol. Biol.* 2023), the human Xi in hTERT-RPE1 cells did not (Supplementary Fig. 7a). This observation led us to hypothesize that the human Xi exhibits less uniform late replication than the mouse Xi.

To test this, we developed a simple metric, the DNA copy number uniformity score, to quantify the consistency of replication status across bins within a chromosome in individual cells (Supplementary Fig. 7b,c). The logic is as follows: chromosomes that replicate uniformly at a given time point should exhibit similar replication status across most bins, resulting in a higher score. For example, in early S-phase (most regions are unreplicated) and late S-phase (most regions are fully replicated), uniformity is high. In contrast, during mid S-phase, when replication is already completed in some bins/regions while others remain unreplicated, uniformity is reduced.

Applying this analysis, we found that the mouse Xi followed the expected pattern: high and stable uniformity during early to mid S phase (up to ~60% replication score), followed by a sharp drop to lower uniformity in mid to late-S (~60–80% replication score), before quickly recovering to high uniformity in late-S phase. By contrast, the human Xi exhibited persistently lower copy number uniformity scores across all stages of S-phase, suggesting that replication on the human Xi is more variable (Supplementary Fig. 7d). We also performed the same analysis on the Xa in both species. In both human and mouse cells, the Xa exhibited somewhat similar DNA copy number uniformity patterns: higher uniformity during early and late S phases, and lower uniformity in mid S, although the pattern was less clear in human cells (Supplementary Fig. 7d).

While this may reflect true biological differences, technical factors such as SNP density or mappability could also contribute to these observations. Nonetheless, these results again underscore the limitations of population-based haplotype-specific BrdU-IP data, which could not clarify these differences. These results highlight the advantage of using scRepli-seq data.”

4. In lines 157-162 on pages 6-7, why are non-poly(A) RNAs detected as efficiently as with scRamDA-seq, but lncRNA detection efficiency is lower? What is the preference of this technique for detecting RNAs? Is the detection efficiency for all such RNAs the same except well-known non-poly(A) RNAs?

Thank you for your comments. We agree that these points require further clarification. To address them, we performed additional analyses and experiments. The new results are presented in Extended Data Fig. 3d,e and Supplementary Fig. 3. Specific responses to your questions are provided below:

In principle, RamDA-seq and scRR-seq-RNA can capture both poly(A)+ and non-poly(A) RNAs, as they utilize a combination of oligo(dT) primers and not-so-random primers for reverse transcription (Hayashi et al., *Nat. Commun.* 2018). Therefore, this approach enables broad RNA detection, covering both coding and non-coding RNAs.

Non-poly(A) RNAs include a diverse group of transcripts, such as certain histone mRNAs and lncRNAs. As shown in Fig. 1f, scRR-seq-RNA could detect well-known non-poly(A) RNAs with efficiency comparable to scRamDA-seq. We reasoned that this is because these transcripts tend to be relatively abundant and stable (e.g., 7 out of 9 genes shown in Fig. 1f have $\log_{10}(\text{median TPM}+1) > 2.19$; Table 2 for Reviewers). This abundance likely facilitates their reverse transcription and capture.

Table 2 for Reviewers: Gene expression level of non-poly(A) RNAs shown in Fig. 1f.

Gene name	median TPM	mean TPM	log10(median TPM+1)	log10(mean TPM+1)
RN7SK	2897.519689	3346.284777	3.462176255	3.448014503
RN7SL1	2219.53313	3274.988533	3.346457257	3.397888306
MALAT1	1273.532764	1666.010486	3.105351004	3.154153421
RNU4-1	1216.633423	1244.052242	3.085516561	3.038415483
NEAT1	502.5591023	561.3910335	2.702050451	2.597358058
RNU1-3	170.4486681	225.4919425	2.234134116	2.26353915
RMRP	154.3924839	170.9418589	2.191430009	2.180237742
RPPH1	53.36841365	62.4031994	1.735346661	1.652255202
TERC	1.542916624	2.845900149	0.405332121	0.407980159

Nonetheless, as the reviewer mentioned, many lncRNAs were not efficiently detected by scRR-seq-RNA. One reason is that they generally have low expression levels. As shown in Extended Data Fig. 3a, genes detected only in scRamDA-seq but not scRR-seq show lower expression levels than those that can be detected by both methods. However, we hypothesized that other factors, such as subcellular localization, might also contribute. This is because scRR-seq involves the separation of DNA and RNA before profiling. RNAs associated with chromatin or retained in the nucleus (i.e., on beads) may be underrepresented in the cytoplasmic (supernatant) fraction. To test this, we utilized published RNA localization data from several human cell lines (Mas-Ponte et al., *RNA*. 2017) to examine the potential subcellular localization of RNAs detected only by RamDA-seq and those detected by both methods. Indeed, we found that RNAs that were not detected by scRR-seq are more likely to show nuclear localization (Extended Data Fig. 3d,e).

To confirm this further, we performed additional experiments using RPE1 cells and the scRR1 protocol. After cell lysis, we collected RNA from the supernatant and the bead-bound fractions, which were subjected to scRR-seq-RNA and analyzed separately (Supplementary Fig. 3a).

Gene expression quantification by RSEM detected over 15,000 expressed genes (TPM>0) per cell in both supernatant-derived (cytoplasmic) and bead-derived (nuclear) samples (Supplementary Fig. 3b), with a slightly higher number observed in the bead-derived (nuclear) fraction. This indicated that certain RNAs were more enriched in the bead (nuclear) fraction. Clustering analysis identified a group of genes that shows a higher expression level in the bead fraction than the supernatant (Supplementary Fig. 3c, cluster 2). This group was significantly enriched for lncRNAs (Supplementary Fig. 3d). These results indicate that a subset of lncRNAs preferentially associates with the bead (nuclear) fraction.

Again, using published RNA localization data (Mas-Ponte et al., *RNA*. 2017), we observed that these bead-enriched RNAs (including lncRNAs) are often found in the nucleus (Supplementary Fig. 3e, cluster 2). We also found that this group of RNAs doesn't show very high expression levels (Supplementary Fig. 3c, cluster 2). These results were consistent with our earlier observation that a subset of genes was detected by scRamDA-Seq but not scRR-seq (Extended Data Fig. 3d,e).

In summary, our findings suggest that the inability of scRR-seq to detect some RNAs, including lncRNAs, is not due to its inability to detect certain types of RNAs. Rather, it is likely due to the characteristics of these RNAs, which tend to be lowly expressed and show nuclear localization. While RamDA-seq is designed to detect all types of RNAs without bias, scRR-seq-RNA detection efficiency varies depending on the RNA characteristics, such as abundance and subcellular localization.

To reflect these things, we revised the main text as follows:

[Pages 6–7]

“Our scRR-seq-RNA detected ~80% of genes and transcripts that were detected by the original scRamDA-seq (Fig. 1e and Extended Data Fig. 2g). scRR-seq-RNA also detected well-known non-poly(A) RNAs as efficiently as scRamDA-seq (Fig. 1f). The remaining ~20% not detected by scRR-seq-RNA were those that show low expression levels by scRamDA-seq and frequently corresponded to long non-coding (lnc)RNAs (Extended Data Fig. 3a–c). Using published RNA localization data²¹, we found that these undetected RNAs tend to be nuclear-localized (Extended Data Fig. 3d,e). This is consistent with the fact that, in scRR-seq-RNA, RNA was collected from the supernatant, which mainly corresponds to the cytoplasmic fraction. We further confirmed that a subset of RNAs, enriched for lncRNAs, remained bound to the beads (Supplementary Fig. 3). These results suggest that RNA abundance and subcellular localization influence detection sensitivity in scRR-seq-RNA. This could potentially explain the decrease in the number of genes detected compared to scRamDA-seq. Overall, however, scRR-seq-RNA data were comparable to scRamDA-seq data obtained individually.”

5. The discussion on the relationship between DNA copy number and gene expression is highly valuable and effectively showcases the capabilities of the authors' method. I have only one minor question here: as replication progresses, does the overall gene expression level in the cell increase? Could the absolute expression of 2-copy genes increase without being reflected in the $[\log_{10}(TPM+1)]$ values?

Thank you for your questions. Below, we responded to these questions one by one as follows:

(5.1) As replication progresses, does the overall gene expression level in the cell increase?

A recent study suggested that RNA content per cell increases by about twofold from the G1 to the G2/M phases (Riba et al., *Nat. Commun.* 2022; <https://doi.org/10.1038/s41467-022-30545-8>; see Fig. 3C in that paper). However, these dynamic changes appear to vary across different cell types. For instance, in mESCs, the increase occurs continuously, whereas in human fibroblasts, RNA levels tend to remain relatively stable from late G1 to early G2/M. Nonetheless, it is unclear whether the overall gene expression levels increase during the S phase.

To address this question, we used the human haploid HAP1 scRR-seq data collected from the whole-S phase. We measured the overall expression level of a gene by counting the number of sequencing reads mapped to the gene in a cell, based on the assumption that read counts reflect the transcription level (Conesa et al., *Genome Biol.* 2016). For a fair comparison, we scaled all scRR-seq-RNA data to 1 million reads per cell, then used featureCounts to quantify the number of reads mapped to each gene. Reads mapped to mitochondrial and ribosomal genes were removed from downstream analysis. To normalize the data, we used total reads from ERCC spike-in controls. These synthetic RNA molecules are added in equal amounts to all cells and serve as stable internal standards unaffected by the cell cycle stage. We then plotted the ERCC-normalized total mapped reads per cell against the percentage replication score (Fig. 3 for Reviewers) and applied linear fitting to capture the overall trend. Our results suggest that the total gene expression levels do not continuously increase during the S phase in HAP1 cells. Rather, they remain relatively stable, with higher cell-to-cell variability during mid-S phase. However, these trends may possibly differ in other cell types.

Fig. 3 for Reviewers. Overall gene expression per cell during S-phase progression.

ERCC-normalized total mapped reads per cell against the percentage replication score in HAP1 cells. Each point represents a single cell. The red line shows the linear fit.

(5.2) Could the absolute expression of 2-copy genes increase without being reflected in the $[\log_{10}(TPM+1)]$ values?

First, we apologize for the mistake in the previous figure label. We did not use TPM for this analysis. Actually, we directly used raw read counts from featureCounts, so the correct label in this figure and related information should have been “ $\log_{10}(\text{mapped reads} + 1)$ ” instead of “ $\log_{10}(\text{TPM} + 1)$.”

To validate this result, we repeated the analysis using the same HAP1 dataset, but this time, we adjusted all samples to the same library size (1 million reads per cell) to ensure a fair comparison of gene expression levels across samples. We examined both raw mapped counts and \log_{10} -normalized values (Fig. 4 for Reviewers). To provide a more detailed view, we categorized gene expression patterns into five groups as shown below (Fig. 4 for Reviewers). We found that ~58% of spliced RNAs had higher median expression in the 2-copy gene group, while ~36% showed higher expression in the 1-copy gene group. However, most of these differences were not statistically significant. Only ~9% of spliced RNAs showed significant differences in gene expression levels between the two groups. For unspliced RNAs, ~48% had higher median expression in the 2-copy gene group, and ~40% showed higher expression in the 1-copy gene group. Similarly, ~7% of unspliced RNAs showed statistically significant differences.

We have updated the manuscript with these new results using $\log_{10}(\text{mapped reads} + 1)$ as shown in Fig. 5b,d, and Supplementary Table 4.

Fig. 4 for Reviewers. Gene expression patterns of replicated (2-copy) and unreplicated (1-copy) genes during mid-S phase in HAP1 cells.

Pie charts show five categories of gene expression patterns. Both mapped reads and $\log_{10}(\text{mapped reads} + 1)$ were used to assess median gene expression levels and perform statistical analysis. The Wilcoxon test was used for statistical comparison. Numbers on the charts indicate the percentage of genes in each category, and the number below each chart shows the total number of informative genes analyzed.

6. It would also be helpful if the authors could discuss the cost and time efficiency of their method compared to other techniques in the Discussion section.

Thank you for this helpful suggestion. We have summarized the cost of scRR-seq in a new Supplementary Table 5. Also, we added a paragraph in the Discussion section to address the cost and time efficiency of our method in comparison with other single-cell multi-omics techniques, as follows:

[Page 17]

“Regarding the overall cost and time efficiency, we believe scRR-seq is comparable to most existing protocols. It typically requires approximately 2–4 days each for DNA-seq and RNA-seq library preparation, which is similar to standard NGS workflows. The scRR-seq protocol is simple and easy, as it utilizes the commercially available RamDA-seq kit for RNA-seq. This simplifies the workflow and reduces variability associated with manual protocols. In terms of cost, our optimized scRR-seq protocol lowers reagent expenses to about \$60 per sample for combined DNA and RNA-seq library preparations (see Supplementary Table 5). In comparison, most other methods use Smart-seq-based technology for the RNA-seq component, which either requires manual protocols or commercial kits that are substantially more expensive (which cost ~\$13 and ~\$80/sample for RNA-seq alone, respectively⁴⁰). Although scRR-seq may not provide a major advantage in processing time or total cost, its simplified workflow makes it more user-friendly and easier to adopt in standard laboratory settings.”

7. Line 260: “We obtained a trajectory of S-phase progression relatively similar to CBMS1 mESCs (Extended Data Fig. 5a)” should instead reference Extended Data Fig. 5d.

Thank you for pointing this out. We have corrected the figure reference as suggested. As we have modified the figures, the sentence now correctly refers to Fig. 3d.

8. In Fig. 4d, is the scRT shown for chr4 or chrX? The title and caption indicate chr4, but the genomic range in the figure is labeled as “chrX:1-60.”

Thank you for pointing this out. We apologize for the mistake in the title and caption, which incorrectly referred to chrX. The scRT data in Fig. 4d is indeed from chr4, and the genomic label should be chr4:1–60.

We have corrected the label in the revised version to accurately reflect that the data shown corresponds to chr4.

Reviewer #3 (Remarks to the Author):

In the manuscript, Poonperm et, al. developed the single-cell Repli-RamDA-seq, also called scRR-seq. As a multi-omics co-assay tool, sRR-seq can be applied to profile high-resolution DNA replication and quantify full-length total RNA within the same cell. Specifically, scRR-seq has shown promising performance regarding the DNA-seq and RNA-seq quality when compared to scRepli-seq, scRamDA-seq, and G&T-seq. The authors have also applied scRR-seq to study the dynamics of the cell cycle on transcriptome and found that S-phase-associated genes are cell type-specific, and a uniform cell cycle marker set can not define cell cycle status accurately. Moreover, they have explored the effect of DNA replication on gene expression. Overall, scRR-seq can be a powerful tool for studying cells from the perspective of genome and transcriptome. I also have a few comments:

We appreciate the reviewer for the positive feedback on our study. We have addressed the reviewer's specific questions and comments in detail below.

Major comments:

1. It is encouraged to release all code for the data analysis of this paper to ensure the result is reproducible. Also, the sequencing data can not be found through GEO.

Thank you for the suggestions. To support reproducibility, the code used for data analysis will be made publicly available on GitHub.

GitHub link: <https://github.com/mcbmieue/scRR-seq>

As the repository is not yet publicly accessible, we have deposited the analysis code on Figshare for your review. It can be accessed using the following link.

Figshare link: <https://figshare.com/s/f7b3fba5caa1657176fd>

We apologize for the issue with accessing our sequencing data. The dataset has been deposited in GEO and can be accessed using the following link and token:

GEO link: <https://www.ncbi.nlm.nih.gov/geo/query/acc.cgi?acc=GSE278959>

Access token: khelqgendqjtm

These details have also been included in the Data Availability section of the revised manuscript.

2. I noticed that you have cited DR-seq without benchmarking with it. Is there any specific reason for that?

Thank you for your question. We cited DR-seq to acknowledge its contribution as a related method, but we did not include a direct comparison with our scRR-seq method for the following reasons:

- **Limited data availability:** DR-seq was performed on two different cell lines: E14 mESCs and the human breast cancer cell line SK-BR-3. Of these, only the E14 mESC data are applicable to our study, as we also used mESCs. However, DR-seq provides DNA data for only three E14 mESC samples, which we considered insufficient for a fair or reliable comparison.

- **Technical differences in the DNA-seq component:** The way DR-seq processes RNA and DNA differs substantially from our method. In DR-seq, RNA and DNA are not physically separated but are instead processed together in a single reaction that is subsequently split for sequencing. This design introduces several analytical challenges distinct from those encountered in our approach. For example:
 - (i) Exclusion of DNA reads: DR-seq has difficulty distinguishing DNA reads that overlap coding regions, as such reads could originate from either cDNA or genomic DNA. To address this, the original authors excluded reads overlapping coding regions from their DNA-seq analysis. Although they noted that these regions account for only ~2% of the genome and do not impact large CNV detection, this exclusion could alter the genome representation and complicate downstream comparison for high-resolution copy number analysis that we use to assess DNA replication status.

 - (ii) Unequal genomic binning: Due to the masking of coding regions, DNA reads in DR-seq are assigned to genomic bins of unequal size. This is incompatible with our scRR-seq-DNA pipeline, which relies on consistent, high-resolution binning to accurately analyze replication state and CNVs.

 - (iii) Amplification biases: DR-seq uses quasilinear amplification, which introduces specific biases. Although these biases were corrected using a custom coverage-based model developed by the DR-seq authors, such corrections are not directly transferable to our data or analytical framework.

As an example, we plotted read coverage of DNA-seq from DR-seq (DR-seq-DNA) in E14 mESCs along chromosome 5 and compared with reads per million (rpm) of our scRR-seq-DNA (Fig. 5a,b

for Reviewers). Since no cell-cycle information is available for the DR-seq cells, we assumed that they were in S-phase, as we observed higher and lower coverage reads in early-replicating and late-replicating regions, respectively, (Fig. 5b,c for Reviewers; upper panel shows population-based mESC RT profile). However, the read distribution within those regions is uneven (Fig. 5b for Reviewers). This unevenness may result from: (1) a mixture of replicated (high copy) and unreplicated (low copy) DNA within the same region; (2) exclusion of reads overlapping with cDNA; or (3) normalization procedures correcting for amplification bias. In contrast, our scRR-seq-DNA from S-phase cells shows a much more uniform read distribution, with higher reads in early-replicating regions and lower reads in late-replicating regions (Fig. 5b,d for Reviewers). As a result, when we performed log₂median transformation on our scRR-seq-DNA data, the profiles closely resembled the RT profile of population mESCs (Fig. 5a,b for Reviewers; upper panel shows population-based mESC RT profile), while DR-seq data show less clear results. To quantify the variability of data, we calculated the median absolute deviation (MAD) across the genome (Fig. 5e for Reviewers), and across early- and late-replicating regions (Fig. 5f for Reviewers). DR-seq-DNA showed higher MAD scores in all regions analyzed, indicating high variability and less uniform coverage.

In summary, while DR-seq may be applicable for general CNV detection, we believe it is less appropriate for analyzing high-resolution replication timing based on copy number analysis. Taken together, we considered that comparing the DR-seq and scRR-seq directly may give a misleading impression, rather than a fair evaluation.

Fig. 5 for Reviewers. DNA-seq read distribution of DR-seq and scRR-seq in mouse ESCs.

a) Read coverage along chromosome 5 (chr5) from DR-seq-DNA in three E14 mESCs and read count per million (rpm) from scRR-seq-DNA in three CBMS1 mESCs. The top panel shows BrdU-IP Repli-seq (population-based) RT profiles from JB4 mESCs (Poonperm et al., *Nat. Struct. Mol. Biol.* 2023) for a comparison. Yellow and blue represent early and late RT domains, respectively. Log2median profiles from scRR-seq-DNA closely resemble the population-based RT profiles, whereas those from DR-seq-DNA appear less defined. b) Zoomed-in view of the

region shown in a). e) MAD scores calculated for DR-seq-DNA and scRR-seq-DNA data across the whole genome for each single cell. f) MAD score calculated for DR-seq-DNA and scRR-seq-DNA across early- and late-replicating regions for each single cell. Bins with BrdU-IP Repli-seq RT value > 0.25 were classified as Early-RT, and $RT < -0.25$ as Late-RT.

3. As far as I know, it will largely increase the scope of scRR-seq when you apply it to cancer studies, such as the relationship between CNV, mutations, and gene expression. Will it be able to provide one additional figure for such a case study?

We greatly appreciate the reviewer's insightful suggestion to explore the application of scRR-seq in cancer research. We fully agree that combining scRR-seq with CNV, mutation, and gene expression analysis in cancer cells would significantly enhance the impact and utility of this technology. However, due to limitations in sample availability and resources, we are currently not in a position to include a cancer-focused case study in the present work. While we understand that such an application would broaden our work, we feel that it would be a bit beyond the intended scope of this manuscript.

In response to this suggestion, we have added a perspective on how scRR-seq could be applied to cancer biology in the Discussion section as follows:

[Page 17]

“Since the regulation of gene expression associated with CNVs is complex and not fully understood³⁸, scRR-seq offers a unique opportunity to explore this relationship at the single-cell level. **In particular, scRR-seq could be highly useful in cancer research, where gene expression and CNVs vary between cells and influence tumor progression and therapeutic response³⁹.** Furthermore, we recently demonstrated that scRepli-seq can identify CNVs in mouse embryos at the single-cell level, regardless of whether cells are in G1 or S-phase¹⁶. This suggests that scRR-seq could be extended to explore gene expression and CNV associations in S-phase cells, which would be particularly valuable for analyzing rare or limited samples, for instance, single cells derived from tissues *in vivo* **or patient samples.**”

4. The comparison of identifying S phase cells between the Seurat gene set and SRR-seq is unfair. Will it also be able to generate the data from other known phages as ground truth and build a model to predict cell-cycle status using scRR-seq? The model can be very simple, such as a linear model. Then it is possible to compare the accuracy of the two methods for identifying cell cycle status.

Thank you for your valuable feedback. First, we apologize for any confusion caused by our original explanation. Our primary focus in this study is not on comprehensive cell cycle classification, but rather on capturing gene expression dynamics during S-phase progression using scRR-seq. This is because one of the key advantages of scRR-seq is its ability to simultaneously assess transcriptional and replication dynamics within S-phase, which are difficult using conventional gene expression-based methods.

We agree with the reviewer that the original comparison between the Seurat cell-cycle gene set and scRR-seq S-phase progression markers was not fully fair. The Seurat gene set is meant to broadly classify cells into G1, S, and G2/M phases using genes that are specifically expressed in S- and G2/M-phases. In contrast, our S-phase progression markers were developed based on data derived from only S-phase cells and are intended to resolve cell states only within the S phase, not across the entire cell cycle. These differences in purpose and scope have introduced bias in the comparison. In addition, our original comparison may have also created the misleading impression that our newly identified markers were intended for full cell cycle classification, and vice versa. To address these issues, we have removed the comparison between the Seurat cell-cycle gene set and scRR-seq's S-phase progression markers (original Fig. 3e,f, middle panels) from the revised manuscript.

We also appreciate the suggestion to build a predictive model for cell-cycle phases using scRR-seq data. However, one limitation of scRR-seq-DNA (and scRepli-seq), as currently implemented, is that we cannot clearly distinguish between G1 and G2/M phases, due to the challenge of resolving these states based solely on DNA copy number. While our HMM-based DNA replication analysis is well-suited for detecting progression through S-phase, we cannot confidently assign G1 or G2/M phases. For instance, as noted in our responses to Reviewer 2's comment #2, we cannot distinguish whether the cells from embryo #1 are in G1 or G2/M phase. As such, building a full cell-cycle phase prediction model based solely on scRR-seq would be limited.

In response to your comments, we have revised the manuscript to tone down our claims about full cell-cycle identification and instead emphasize the strength of our method in capturing detailed S-phase progression and identifying novel S-phase progression markers, which, to our knowledge,

have not been reported so far. We have rewritten the original section titled “*scRR-seq reveals cell-cycle gene expression dynamics at high resolution*” and divided it into two sub-sections. We first highlight the ability of scRR-seq to identify new markers associated with S-phase progression. Then, we discuss the limitations of existing cell-cycle markers and assignment methods. We hope this revised version more accurately reflects the scope and strengths of our method.

The revised text is as follows:

[Pages 9–11]

scRR-seq captures gene expression dynamics and novel markers during S-phase progression

As scRT data allows one to tell the S-phase timepoint of each cell (Supplementary Fig. 5), we reasoned that scRR-seq data should reveal gene expression dynamics during **S-phase progression** at an unprecedented temporal resolution **and may help identify S-phase progression markers, which have never been reported before**. Therefore, we generated scRR-seq data from hTERT-RPE1 cells and CBMS1 mESCs throughout the S-phase. First, we used scRR-seq-DNA to generate a ‘whole-S’ RT profile, which showed that we successfully captured the entire S-phase of hTERT-RPE1 cells (Fig. 3a) and CBMS1 mESCs (Fig. 3b and Extended Data Fig. 5). The whole-S RT profile of CBMS1 mESCs closely resembled the profile obtained previously by scRepli-seq¹³ (Extended Data Fig. 6a). Then, we used our hTERT-RPE1 and CBMS1 mESC scRR-seq-RNA data to find novel S-phase progression marker candidates. We sorted cells based on their percentage replication scores and searched for genes that showed significant gene expression changes during S-phase progression. Using the generalized additive model (GAM) fitting, we identified 52 significant dynamic genes (FDR<0.05) in hTERT-RPE1 and 55 genes in CBMS1 mESCs (FDR<0.01) and divided them into three groups based on their expression patterns (Extended Data Fig. 6b,c and Supplementary Table 2). Gene Ontology (GO) analysis revealed that these newly identified genes were significantly overrepresented in processes related to cell division (Supplementary Table 2). Only a few genes were shared between the two cell lines (Supplementary Table 2).

Utilizing these newly identified genes for diffusion map analysis (DMA), we observed a clear trajectory of S-phase progression (Fig. 3c). However, these new human **S-phase progression** markers were unable to reveal the S-phase progression trajectory in mice and vice versa, suggesting the species- or cell-type-specificity of markers (Fig. 3c, **Supplementary Fig. 6a–e**). Lastly, we applied these newly identified S-phase progression markers of CBMS1 to another mESC scRNA-seq dataset¹⁴ to assess their reproducibility. We obtained a trajectory of S-phase progression relatively similar to that of CBMS1 mESCs (Fig. 3d), confirming the robustness of these markers. Taken

together, our results demonstrate that scRR-seq enables the comprehensive analysis of gene expression dynamics during **S-phase progression** at an unprecedented temporal resolution. Moreover, it also allows the identification of a novel set of S-phase progression markers for a given cell type and species.

scRR-seq reveals a limitation of existing cell-cycle stage assignment

Cell-cycle markers are commonly used to assign cell-cycle stages to scRNA-seq datasets. For example, Seurat²³, a widely used scRNA-seq analysis pipeline, employs a predefined set of S and G2/M markers from human cells²⁴ to classify cells into specific cell-cycle stages. Since our scRR-seq datasets capture both the S-phase state (Supplementary Fig. 5) and gene expression profiles, we leveraged this dual capability to assess the accuracy of existing cell-cycle markers and cell-cycle stage assignments.

We used Seurat and its default cell-cycle marker sets to assign cell-cycle stages to individual cells in our whole-S-derived hTERT-RPE1 and CBMS1 mESCs scRR-seq datasets, based on the scRR-seq-RNA data. We then compared these assignments with the percentage replication scores derived from the corresponding scRR-seq-DNA data.

Unexpectedly, while all cells were in S-phase based on the percentage replication scores, Seurat assigned G1, S, and G2/M phases to these cells (Fig. 3e,f). In hTERT-RPE1 cells, while the S and G2/M phase assignments were relatively reasonable and marked cells with low and high percentage replication scores, respectively, some cells were incorrectly labeled as G1 (Fig. 3e).

In contrast, in CBMS1 mESCs, the majority of cells were assigned as G1 (Fig. 3f). The proportion of cells assigned as S and G2/M was noticeably low, which corresponded to those with low (<40%) and high (>70%) replication scores, respectively (Fig. 3f). Given that Seurat's default markers are derived from human tumor cells²⁴, our findings suggest that cell-cycle markers may be cell-type or species-specific (Supplementary Fig. 6f). Overall, these results underscore a limitation of current markers or methodology for cell-cycle stage assignment, particularly when applied across different cell types or species.

Minor comments:

5. The comparison between scRR-seq to G&T-seq is not fully fair. For example, in the DNA part, scRR-seq-DNA adjusted reads to about 4M/sample. It should also consider cell number and make sure the similar level of mean read count per cell. This is to address my concern about the potential large difference in cell number per sample between SRR-seq and G&T-seq.

Thank you for your suggestions. To address your concern, we improved the comparison between scRR-seq and G&T-seq by separating the G&T-seq samples into two groups based on their initial read counts: one with higher read counts than our samples [referred to as G&T-seq(H)], and another with lower read counts [referred to as G&T-seq(L)] for both DNA-seq and RNA-seq components. We then repeated the comparison, as shown in the **updated Fig. 2b** and **Extended Data Fig. 4**. We found that, regardless of the initial read counts, the DNA-seq component of G&T-seq still did not achieve the same high coverage as our scRR-seq-DNA. Similarly, the RNA-seq results from both G&T-seq(H) and G&T-seq(L) were consistent with each other, showing lower transcript and gene detection compared to scRR3, but comparable to scRR1 in this experiment. These results support our original conclusion that scRR-seq shows better performance over G&T-seq under the tested conditions.

6. Please also provide additional support for your saying that cell-cycle markers are cell-type-specific. In your provided evidence, the difference may be introduced by different species.

Thank you for this suggestion. We agree that species differences could have contributed to the observed variation, and we appreciate the opportunity to provide additional support for our claim.

We would like to clarify that our current focus is specifically on S-phase progression markers, rather than general cell-cycle markers across all phases. To address your concern, we have now included new data on S-phase progression markers in human-derived HAP1 cells, which are shown in the new Supplementary Fig. 6a–e. These new results support the idea that S-phase progression markers can vary between different cell types, even within the same species, and also suggest that S-phase progression markers are cell-type-specific.

In addition, we also used Seurat and its default cell-cycle markers and applied them to our HAP1 scRR-seq-RNA data (new Supplementary Fig. 6f). Similar to what was observed in hTERT-RPE1 (Fig. 3e), despite HAP1 cells being in S-phase based on the percentage replication scores, Seurat

assigned them to G1, S, and G2/M phases. While the S and G2/M phase assignments were relatively reasonable and marked cells with low and high percentage replication scores, respectively, some cells were assigned as G1. Taken together, these results suggest that cell-cycle markers may be cell-type or species-specific.

7. It is not surprising that DNA copy numbers can not influence gene expression in normal cells. As I have mentioned in major comments, the relationship between CNV and gene expression in cancer cells can be more interesting.

Thank you for your comment. As mentioned in our response to comment 3, we agree that the relationship between CNVs and gene expression in cancer cells is particularly interesting, and we also recognize the potential of scRR-seq in this area. Therefore, we have expanded the Discussion section to highlight its possible application to cancer research.

Reviewer #4 (Remarks to the Author):

We appreciate the reviewer for their time and thoughtful contribution to the review process.